# Mitochondria are required for pro-ageing features of the senescent phenotype

Clara Correia-Melo[1,2], Francisco DM Marques[1], Rhys Anderson[1], Graeme Hewitt[1], Rachael Hewitt[3], John Cole[3], Bernadette M Carroll[1], Satomi Miwa[1], Jodie Birch[1], Alina Merz[1], Michael D Rushton[1], Michelle Charles[1], Diana Jurk[1], Stephen WG Tait[3], Rafal Czapiewski[1], Laura Greaves[4], Glyn Nelson[1], Mohammad Bohlooly-Y[5], Sergio Rodriguez-Cuenca[6], Antonio Vidal-Puig[6], Derek Mann[7], Gabriele Saretzki[1], Giovanni Quarato[8], Douglas R Green[8], Peter D Adams[3], Thomas von Zglinicki[1], Viktor I Korolchuk[1] & João F Passos[1,*]

## Abstract

Cell senescence is an important tumour suppressor mechanism and driver of ageing. Both functions are dependent on the development of the senescent phenotype, which involves an overproduction of pro-inflammatory and pro-oxidant signals. However, the exact mechanisms regulating these phenotypes remain poorly understood. Here, we show the critical role of mitochondria in cellular senescence. In multiple models of senescence, absence of mitochondria reduced a spectrum of senescence effectors and phenotypes while preserving ATP production via enhanced glycolysis. Global transcriptomic analysis by RNA sequencing revealed that a vast number of senescent-associated changes are dependent on mitochondria, particularly the pro-inflammatory phenotype. Mechanistically, we show that the ATM, Akt and mTORC1 phosphorylation cascade integrates signals from the DNA damage response (DDR) towards PGC-1β-dependent mitochondrial biogenesis, contributing to a ROS-mediated activation of the DDR and cell cycle arrest. Finally, we demonstrate that the reduction in mitochondrial content *in vivo*, by either mTORC1 inhibition or PGC-1β deletion, prevents senescence in the ageing mouse liver. Our results suggest that mitochondria are a candidate target for interventions to reduce the deleterious impact of senescence in ageing tissues.

**Keywords** ageing; inflammation; mitochondria; mTOR; senescence
**Subject Categories** Ageing; Cell Cycle; Metabolism
**The EMBO Journal (2016) 35: 724–742**

See also: **N Herranz & J Gil** (April 2016)

## Introduction

Cellular senescence, the irreversible loss of proliferative potential in somatic cells, has been shown to not only play a major role in tumour suppression, but also to contribute to age-related tissue degeneration (Baker *et al*, 2011). While senescence can stably repress cancer by irreversibly arresting damaged cells, senescent cells have been shown to be detrimental in tissues, mainly by developing a complex senescence-associated secretory phenotype (SASP) (Coppé *et al*, 2008; Kuilman & Peeper, 2009) and producing elevated levels of reactive oxygen species (ROS) (Passos *et al*, 2010). These factors not only contribute to the maintenance of the senescence arrest via self-amplifying positive feedback loops (Acosta *et al*, 2008; Kuilman *et al*, 2008; Passos *et al*, 2010), but can also induce senescence in healthy cells in a paracrine fashion (Nelson *et al*, 2012; Acosta *et al*, 2013).

Mitochondria play an essential role in energy generation, cell signalling, differentiation, death and senescence in eukaryotic cells. Mitochondria's role in cellular senescence is widely associated with their significant role as generators of ROS-mediated random molecular damage. ROS have been shown to play important roles in senescence by inducing genomic damage (Parrinello *et al*, 2003), accelerating telomere shortening (von Zglinicki, 2002) and acting as drivers of signalling networks important for the maintenance of the senescent phenotype (Passos *et al*, 2010). However, other reports have also highlighted the importance of non-mitochondrial ROS sources (Takahashi *et al*, 2006), redox stress (Kaplon *et al*, 2013) or

1   Institute for Cell and Molecular Biosciences, Campus for Ageing and Vitality, Newcastle University Institute for Ageing, Newcastle University, Newcastle upon Tyne, UK
2   GABBA Program, Abel Salazar Biomedical Sciences Institute, University of Porto, Porto, Portugal
3   Institute of Cancer Sciences, CR-UK Beatson Institute, University of Glasgow, Glasgow, UK
4   Wellcome Trust Centre for Mitochondrial Research, Newcastle University Centre for Brain Ageing and Vitality, Newcastle University, Newcastle upon Tyne, UK
5   Transgenic RAD, Discovery Sciences, AstraZeneca, Mölndal, Sweden
6   Metabolic Research Laboratories, Wellcome Trust-MRC Institute of Metabolic Science, Addenbrooke's Hospital, University of Cambridge, Cambridge, UK
7   Faculty of Medical Sciences, Institute of Cellular Medicine, Newcastle University, Newcastle upon Tyne, UK
8   Department of Immunology, St. Jude Children's Research Hospital, Memphis, TN, USA
    *Corresponding author. Tel: +44 191 248 1222; Fax: +44 191 248 1101; E-mail: joao.passos@ncl.ac.uk

   

deficits in antioxidant defence (Blander *et al*, 2003) in the development of senescence. Therefore, it remains unknown whether mitochondria are truly necessary for senescence.

To understand to what extent mitochondria are involved in senescence, we took advantage of a recently identified process whereby the depolarization of mitochondria by uncouplers, such as CCCP, targets the ubiquitin ligase Parkin to mitochondria and promotes their degradation via the proteasome and autophagy, resulting in complete elimination of the mitochondrial compartment (Narendra *et al*, 2008; Geisler *et al*, 2010). We found that widespread targeted mitochondria depletion abrogates the development of commonly described features of cellular senescence, such as the pro-inflammatory and pro-oxidant phenotype and expression of the cyclin-dependent kinase inhibitors p21 and p16, while preserving the cell cycle arrest and therefore the tumour suppressor properties of these cells. Furthermore, we delineate a mechanism by which the DNA damage response (DDR), a major driver of senescence, interacts with mitochondria to develop the senescent phenotype. We demonstrate that the ATM, Akt and mTORC1 phosphorylation cascade plays a key role as an effector of the DDR by promoting PGC-1β-dependent mitochondrial biogenesis and cellular senescence *in vitro* and *in vivo*.

## Results

### Mitochondria are key factors in the development of pro-ageing features of senescence

To determine the importance of mitochondria for the establishment and maintenance of cellular senescence, human MRC5 fibroblasts were stably transduced with YFP-Parkin and senescence was induced by X-ray irradiation (20 Gy) (Fig 1A). Parkin-mediated mitochondrial clearance was confirmed by the absence of mitochondrial proteins (Fig 1B), the lack of mitochondrial respiration and insensitivity to electron transport chain-inhibiting drugs (Fig 1C) and the absence of intact mitochondria by 3D electron microscopy (Fig 1D and Movies EV1 and EV2). This was observed not only immediately after CCCP treatment, but also at later time points (up to 16 days) after CCCP had been removed (Fig EV1A and B).

Elimination of mitochondria upon the induction of senescence impacted on the development of specific effectors and phenotypes of senescence. Mitochondria-depleted cells showed substantially reduced cell size, Sen-β-Gal activity, the formation of heterochromatin foci, ROS generation and expression of the cyclin-dependent kinase inhibitors p21 and p16 (Fig 1E–G). By conducting a cytokine array, we found that the secretion of major SASP factors such as IL-6, IL-8, GRO and MCP-1 was drastically reduced following mitochondrial clearance (Fig 1H). Independent ELISAs confirmed that the secretion of the SASP components IL-6 and IL-8 was barely detectable in the culture media of mitochondria-depleted senescent cells (Fig EV1C). Similar results were observed in rho0 cells, which are depleted of mtDNA and possess few petite mitochondria. Rho0 cells produced reduced levels of ROS and the pro-inflammatory cytokine IL-6 in both basal conditions and during X-ray-induced senescence (Fig EV1D).

Live-cell imaging of mitochondria-depleted cells for over 2 weeks revealed cells displaying morphological features of non-senescent fibroblasts (Movies EV3 and EV4). Mitochondria-depleted cells, despite showing a slight increase in proliferation until 10 days, as shown by a small rise in the percentage of Ki67-positive cells and population doublings, had an extremely slow division rate (around 0.3 times that of young proliferating cells) and eventually stopped dividing at 20 days post-induction of senescence (Fig EV1E). Together with these observations, we found that mTOR activity, a master regulator of cell growth and division, was decreased in senescent mitochondria-depleted cells (Fig EV1F). Similar results were observed in young proliferating cells, where the depletion of mitochondria resulted in decreased mTOR activity and impairment of cell proliferation (Fig EV1G and H). Nevertheless, the cell cycle inhibition was not accompanied by the development of senescence-associated alterations (Fig EV1H) or decreased ATP levels in the young mitochondria-depleted cells (Fig EV1L).

In order to further understand the impact of mitochondria on senescence, we performed RNA-seq analysis on proliferating (Prol), senescent (Sen) and mitochondria-depleted senescent cells (MDSen). Analysis of all genes that are differentially expressed between Prol, Sen and MDSen showed that there was a tendency for genes down-regulated in senescence to be up-regulated upon mitochondrial depletion and for genes up-regulated in senescence to

---

**Figure 1.  Mitochondria are key factors for the development of pro-ageing features of cellular senescence.**

A   Scheme illustrating the experimental design: Parkin-expressing and control MRC5 fibroblasts were irradiated with 20-Gy X-ray and 2 days after were treated with 12.5 μM CCCP for 48 h. At day 4 after irradiation, CCCP was removed and cells were kept in culture with normal serum-supplemented medium. Cells were harvested at days 10 or 20 after irradiation (6 or 16 days after CCCP treatment, respectively) for senescent phenotypes analysis.

B   Western blots showing the absence of mitochondrial proteins from the different mitochondrial complexes: NDUFB8 (complex I), SDHA (complex II), UQCRC2 (complex III) and MTCO-1 (complex IV), in senescent (10 days after 20-Gy X-ray) control and Parkin-expressing MRC5 fibroblasts. Data are representative of three independent experiments.

C   Cellular oxygen consumption rates (OCR) in senescent (10 days after 20-Gy X-ray) control and Parkin-expressing MRC5 fibroblasts. Data were obtained using the Seahorse XF24 analyzer and show mean ± SD *n* = 4 technical repeats (representative of two independent experiments).

D   Representative 3D EM pictures of senescent (20 days after 20-Gy X-ray) Parkin-expressing MRC5 fibroblasts with or without CCCP. Mitochondria are in red; the nucleus in blue.

E   Representative images of cell size, Sen-β-Gal activity (blue cytoplasmic staining), macro-H2A foci and SDHA immunofluorescence in proliferating and senescent (10 days after 20-Gy X-ray) Parkin-expressing MRC5 fibroblasts. Scale bar = 10 μm.

F   Quantification of ROS levels (DHE fluorescence), Sen-β-Gal-positive cells and senescence-associated heterochromatin foci (SAHF) observed by DAPI in proliferating and senescent (10 days after 20-Gy X-ray) control and Parkin-expressing MRC5 fibroblasts. Data are mean ± SEM of *n* = 3 independent experiments; asterisks denote a statistical significance at *P* < 0.05 using one-way ANOVA.

G   Representative Western blots showing p21 and p16 expression in proliferating and senescent (10 days after 20-Gy X-ray) control and Parkin-expressing MRC5 fibroblasts. Data are representative of 3 independent experiments.

H   Secreted protein measured in a cytokine array (RayBiotech) of a variety of inflammatory proteins in proliferating and senescent (10 days after 20-Gy X-ray) Parkin-expressing MRC5 fibroblasts. Data are mean ± SEM of *n* = 3 independent experiments. Asterisks denote a statistical significance at *P* < 0.05 using one-way ANOVA.

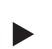

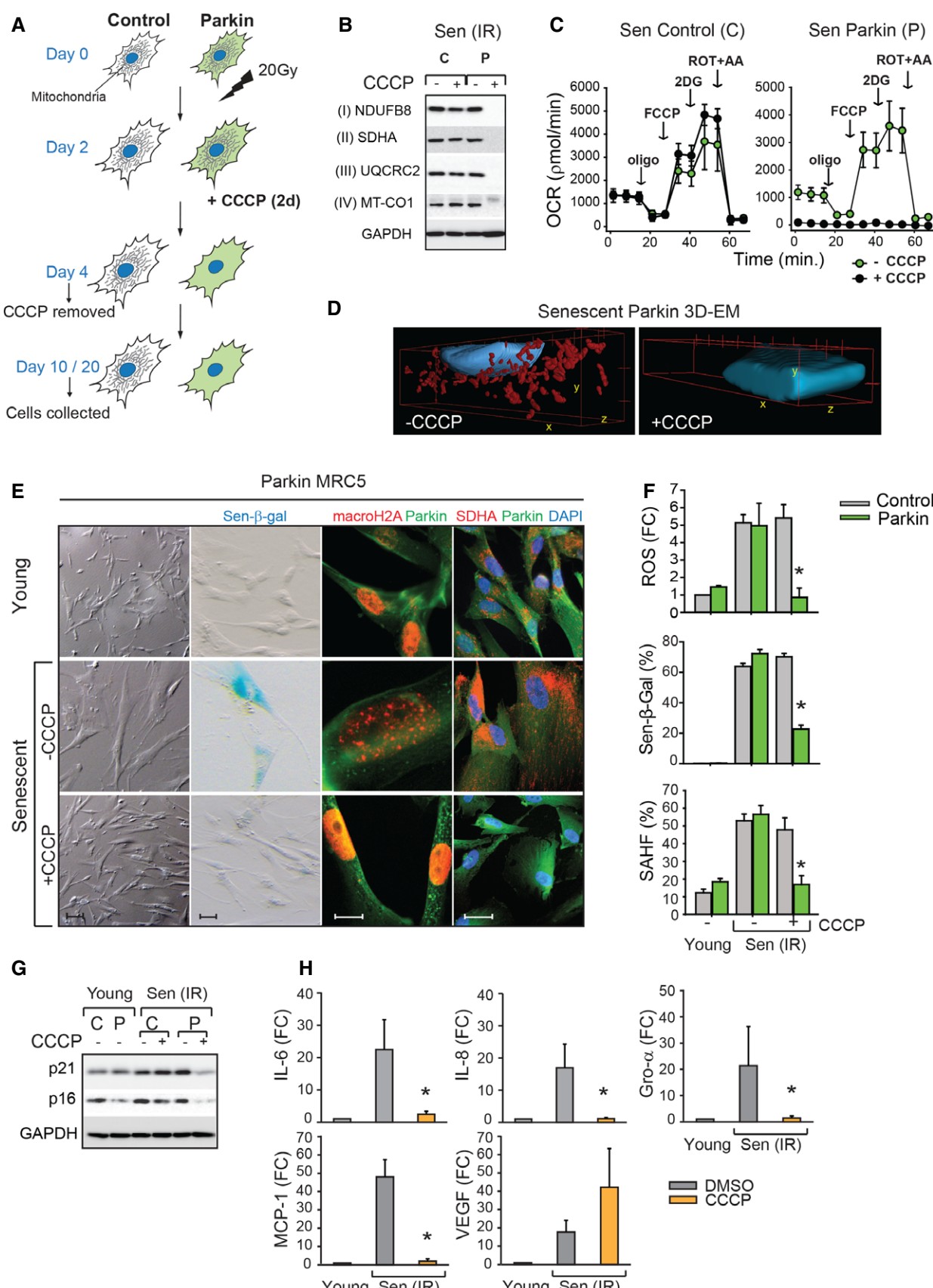

Figure 1.

**A**     All differentially expressed genes (FDR<=5%, absolute fold >=2) between proliferating and senescent cells

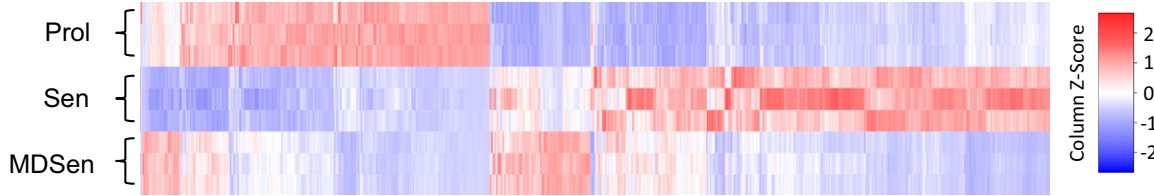

MDSen: **M**itochondrial **D**epleted **Sen**escent cells

**B**

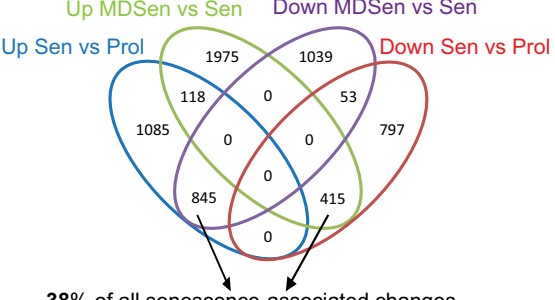

**38**% of all senescence-associated changes
are reversed upon mitochondrial clearance

**C**

10 most significantly enriched gene ontologies for
senescence-associated genes that are **fully reversed** in MD Sen

| Term Name | N of Genes | Fold Enrichment | FDR |
|---|---|---|---|
| response to wounding | 32 | 2.21 | 0.09% |
| cellular chemical homeostasis | 25 | 2.41 | 0.21% |
| cellular homeostasis | 28 | 2.20 | 0.34% |
| female pregnancy | 12 | 3.99 | 0.35% |
| inflammatory response | 22 | 2.47 | 0.41% |
| cellular ion homeostasis | 24 | 2.35 | 0.43% |
| homeostatic process | 38 | 1.85 | 0.62% |
| chemical homeostasis | 29 | 2.07 | 0.66% |
| biological adhesion | 35 | 1.83 | 1.40% |
| cell adhesion | 35 | 1.83 | 1.40% |

**D**

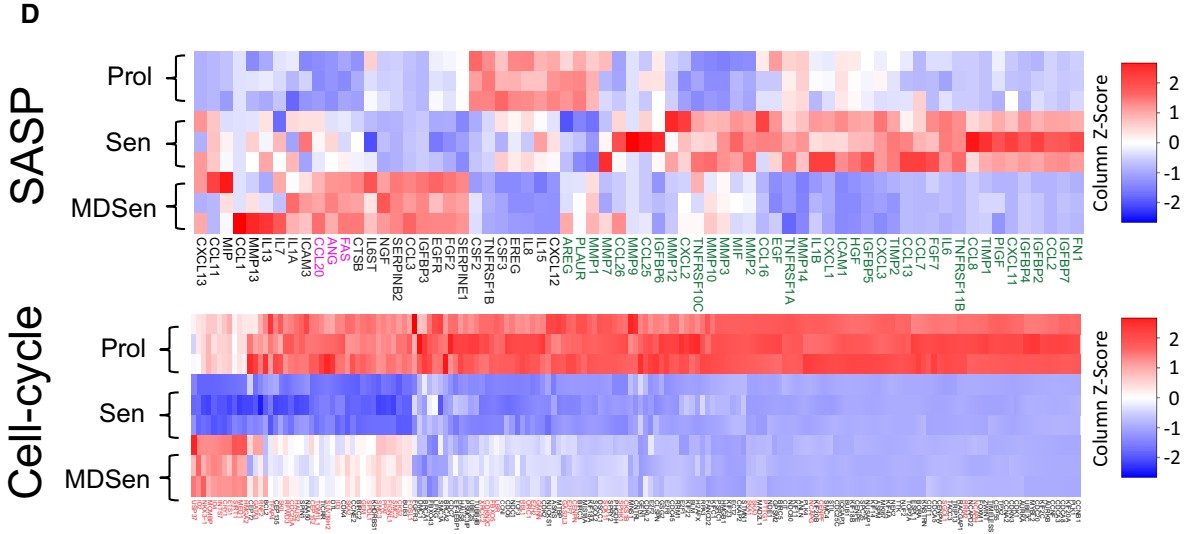

**Figure 2.  Global transcriptomic analysis reveals that mitochondria are significant contributors to the development of the senescent phenotype, particularly the SASP.**

A   Column clustered heatmap showing all genes that are differentially expressed between proliferating (Prol), senescent (Sen) and mitochondria-depleted senescent (MDSen) Parkin-expressing MRC5 fibroblasts (FDR ≤ 5%, absolute fold ≥ 2). Genes are by column and samples by row. The colour intensity represents column Z-score, where red indicates highly and blue lowly expressed.

B   Venn diagram showing the overlap of the differentially expressed genes (FDR ≤ 5%, absolute fold ≥ 2) between proliferating and senescent cells and between senescent and MD senescent cells. Direction of change is indicated.

C   Table of the 10 most significantly enriched (FDR ≤ 5%) gene ontologies for the "fully reversed" genes, as defined by the tool DAVID. Ontologies that are amongst the 10 most significantly enriched for genes that are significantly up-regulated between proliferating and senescent cells are indicated in red.

D   Column clustered heatmaps of all SASP genes and cell cycle genes that are differently expressed in senescence; genes are by column and samples by row. The colour intensity represents column Z-score, where red indicates highly and blue lowly expressed. SASP heatmap: genes that are labelled in green represent the 60% that are fully reversed and genes labelled in pink represent the 5% that became exacerbated upon mitochondria clearance. Cell cycle heatmap: genes labelled in red are significantly changed upon mitochondria clearance.

be down-regulated upon mitochondrial depletion (Fig 2A). We observed that 41% of genes up-regulated in senescence are down-regulated upon mitochondrial depletion (while only 6% are up-regulated) and 33% of genes down-regulated in senescence are up-regulated upon mitochondrial depletion (while only 4% are down-regulated). In total, 38% of all senescent-associated changes are reversed upon mitochondrial depletion (Fig 2B). Analysis of the 10 most significantly enriched (FDR ≤ 5%) Gene Ontologies (GO) for the "fully reversed" genes, as defined by the tool DAVID include "response to wounding", "inflammatory response" and "biological and cell adhesion" which comprise many of the SASP genes (Fig 2C). In fact, by analysing specifically SASP components (Coppé *et al*, 2008), we found that the vast majority was down-regulated at the mRNA level in mitochondria-depleted cells. 60% of the SASP genes which are significantly different between proliferating and senescent were reversed upon mitochondrial depletion, whereas only 5% were exacerbated (Fig 2D). In contrast, analysis of genes belonging to the GO term "cell cycle" showed generally no rescue following mitochondrial clearance, with only a small subset of cell cycle genes increasing at relatively low magnitude (Fig 2D).

One possibility to explain these observations would be that the absence of mitochondria significantly impairs the cell's energy production capacity, thereby precluding the execution of the senescence programme. In order to test this hypothesis, we analysed the bioenergetics status of mitochondria-depleted cells. Analysis of the extracellular acidification rate (ECAR) (Fig EV2A) confirms that mitochondria-depleted senescent cells have a substantially increased glycolytic rate when compared to controls; the addition of the glucose uptake inhibitor 2-deoxyglucose (2DG) has a considerably stronger effect in ECAR of mitochondria-depleted cells than in controls (Fig EV2A). Furthermore, glycolysis-dependent ATP production rate is greater in mitochondria-depleted senescent cells than the ATP production rate driven collectively by both glycolysis and OXPHOS in controls (Fig EV2B). Subsequent measurements of steady-state cellular ATP levels confirmed that these were elevated in mitochondria-depleted cells when compared to senescent controls (Fig EV2C). Consistent with a reliance on glycolysis, glycolysis-associated genes were generally up-regulated in mitochondrial-depleted senescent cells (Fig EV2D). Together, these data suggest that the failure to execute the senescence programme is not dependent on ATP levels.

We next investigated whether mitochondrial clearance impacted on other models of cellular senescence. Similar to X-ray-induced senescence, clearance of mitochondria during oxidative stress-induced senescence (Fig EV3A–C), oncogene-induced senescence (Fig EV3D–F) and replicative senescence (Fig EV3G–J) reduced markers of senescence.

Altogether, these data demonstrated that mitochondria are key factors for the pro-oxidant and pro-inflammatory phenotypes. We next sought to explore the mechanisms that regulate mitochondria in cellular senescence.

## PGC-1β-dependent mitochondrial biogenesis downstream of a DDR modulates cellular senescence

Activation of a DDR is a prominent initiator of senescence during stress-induced (Passos *et al*, 2010), replicative (d'Adda di Fagagna *et al*, 2003) and oncogene-induced senescence (OIS) (Suram *et al*,

2012). In order to understand mitochondria regulation during senescence, we tested the impact of DDR activators on mitochondrial content in a variety of human and mice cell types. DDR activators included X-ray irradiation, oxidative stress, DNA-damaging drugs and telomere dysfunction. In all cases, we observed increased mitochondrial mass 2–4 days after DDR activation, suggesting that increased mitochondrial content occurs downstream of a DDR, irrespective of the stimuli (Appendix Fig S1A). Furthermore, mitochondrial mass increased proportionally to the IR dose (Appendix Fig S1B) and directly correlated with DNA damage foci, p21 expression and induction of Sen-β-Gal and inversely with the proliferation marker Ki67 (Appendix Fig S1C). These data indicate that mitochondrial mass increase occurs downstream of a DDR and suggest that it may play a role in the development of cellular senescence.

Corroborating the observation that mitochondrial content increases downstream of a DDR, gene expression analysis showed that the mitochondrial biogenesis transcriptional co-activators *PGC-1β* and *PGC-1α* were up-regulated upon irradiation (Appendix Fig S1D). We then proceeded to investigate whether the stimulation of mitochondrial biogenesis could play a causative role in cellular senescence. In order to directly evaluate the effect of mitochondrial biogenesis on the induction of senescence, we used mouse embryonic fibroblasts (MEFs) isolated from mice without the mitochondrial biogenesis regulator *PGC1-β*. *PGC-1β*$^{-/-}$ MEFs contained lower mitochondrial mass, mtDNA copy number and mitochondrial proteins expression than wild-type MEFs in both basal conditions and following exposure to senescence-inducing stimuli (Fig EV4A–C).

When cultured at normal atmospheric oxygen (21% $O_2$), MEFs acquire a senescent phenotype after a small number of passages, while when grown at low oxygen (3% $O_2$) MEFs show negligible expression of senescence markers and divide at faster rates (Parrinello *et al*, 2003). We found that at 21% oxygen, the absence of PGC-1β delayed the senescence arrest as shown by colony assays (Fig 3A). In wild-type MEFs, exposure to ambient oxygen or irradiation impacted on the senescent phenotype shown by reduced frequency of Ki67-positive cells, increased Sen-β-Gal activity and the number of 53BP1 foci. All of these phenotypes could be delayed in the absence of PGC-1β (Fig 3B and C). Furthermore, we found that mRNA expression of the senescence-associated genes *p16* and *CXCL1* was significantly reduced in *PGC-1β*$^{-/-}$ MEFs following the induction of senescence by X-ray irradiation (Fig 3D). Similar results were observed in human fibroblasts using a shRNA against PGC-1β (Fig EV4D–F).

In contrast, overexpression of *PGC-1β* (Fig EV4G and H) led to the loss of proliferative capacity and increased Sen-β-Gal activity and frequencies of 53BP1 foci both in proliferating and in IR-induced senescent fibroblasts (Fig 3E). Overexpression of PGC-1β resulted in a significant increase in mitochondrial mass both in basal conditions and during senescence (Figs 3E and EV4I). Together, these data indicate that mitochondrial mass increases following a DDR and establishes a causal relationship between mitochondrial biogenesis and senescence. We then proceeded to explore the signalling pathways linking the DDR to mitochondrial biogenesis.

## ATM, Akt and mTORC1 integrate DDR signalling towards mitochondrial biogenesis during cellular senescence

The mammalian target of rapamycin (mTOR) pathway has been widely implicated in processes governing mitochondrial turnover.

mTOR complex 1 (mTORC1) has been shown to integrate stress signals into the regulation of protein and lipid synthesis and autophagy, all of which are involved in the complex pathways mediating mitochondrial homeostasis (Laplante & Sabatini, 2012).

Following the induction of a DDR using X-ray irradiation, human fibroblasts show a progressive increase in phosphorylation of the mTORC1 target p70S6K (Fig 4A). Consistent with a role for mTORC1 in DDR-dependent mitochondrial protein expression, expression of mitochondrial proteins belonging to the OXPHOS complexes I, II, III and IV (NDUFB8, SDHA, UQCRC2 and MT-CO1) and TOMM20 were significantly reduced by rapamycin treatment, a mTORC1 inhibitor (Fig 4B and Appendix Fig S2A). To test the robustness of our findings, we screened mitochondrial mass as before following treatment with known DDR activators in a variety of human and mouse cell types. In all cases, the increased mitochondrial mass was invariably reduced by rapamycin (Fig 4C). This effect was confirmed at the level of mitochondrial volume fraction and numbers by transmission electron microscopy (T.E.M.), mtDNA copy number (Fig 4D) and live-cell MitoTracker staining in human fibroblasts (Appendix Fig S2B). Comparable effects were observed in MEFs following stress-induced senescence (Appendix Fig S2C). mTORC1 inhibition also repressed the expression of *PGC-1α* and *PGC-1β* genes and the downstream OXPHOS genes *ATP5G1*, *COX5A* and *NDUFS8* (Appendix Fig S2D and E). In contrast, mTORC1 activation via overexpression of constitutively activate mutated Rheb (N153T) (Urano *et al*, 2007) resulted in increased mitochondrial mass in non-stressed cells (Appendix Fig S2F).

Previous studies have suggested links between the DDR and the mTOR signalling pathway potentially via protein kinase B (PKB/Akt) phosphorylation: firstly, Akt has been shown to activate mTORC1 by directly phosphorylating the TSC1/TSC2 complex (Inoki *et al*, 2002) or by dissociation of PRAS40 from the essential mTORC1 component RAPTOR (Thedieck *et al*, 2007). Secondly, Akt has been shown to be a direct phosphorylation target of ATM, an upstream driver of the DDR (Viniegra *et al*, 2005). To test the relation between ATM, Akt and mTORC1, we induced a DDR in human fibroblasts by X-ray irradiation and treated them with an ATM inhibitor. ATM inhibition reduced Akt (S473) and p70S6K (T389) phosphorylation and expression of the cyclin-dependent kinase inhibitor p21 and the mitochondrial protein NDUFB8 following activation of the DDR when compared to controls (Fig 4E and F). We confirmed these observations on human fibroblasts derived from an ataxia telangiectasia (AT) patient, which have impaired ATM signalling (Fig 4E). In support of a pathway involving ATM, Akt and mTORC1 in mitochondrial mass regulation, chemical inhibition of ATM and mTORC1 activity reduced mitochondrial mass with no additive effect when both inhibitors were applied simultaneously (Fig 4G). Our data indicate a novel pathway linking the DDR and mitochondria via the ATM, Akt and mTORC1 phosphorylation cascade and transcriptional activation of mitochondrial biogenesis.

### mTORC1-PGC-1β-dependent mitochondrial biogenesis maintains ROS-driven DNA damage foci (DDF) and contributes to the senescent phenotype

Cumulative data suggest that persistence of a DDR is a key factor in the maintenance of the irreversible arrest observed during cellular senescence. We have previously reported that a significant fraction of the DNA damage foci (DDF) remain unrepaired during cellular senescence and may be a consequence of ROS-generated DNA damage (Passos *et al*, 2010). For these reasons, we decided to investigate the links between mTORC1, mitochondrial biogenesis and the stabilization of a DDR.

mTORC1 inhibition, by rapamycin supplementation, following X-ray irradiation decreased DDF in human and mouse fibroblasts (Fig EV5A and B). Similar effects were observed when exposing cells to DNA-damaging agents such as etoposide, neocarzinostatin (NCS) and hydrogen peroxide ($H_2O_2$) (Fig EV5B). Furthermore, inhibition of mTORC1 resulted in decreased p21 mRNA and protein expression (Fig EV5C).

To test whether ROS and DDF were driven by mTOR, we first knocked down mTOR using siRNA (Fig EV5D) and found that it reduced both ROS and DDF in senescent fibroblasts (Fig 5A). Furthermore, senescent cells supplemented with both rapamycin and the antioxidant NAC showed no cumulative reduction in both ROS and DDF (Fig 5B). Consistent with previous data confirming a central role for a persistent DDR in the development of senescence and the emergence of the SASP (Rodier *et al*, 2009; Hewitt *et al*, 2012), we found that (i) mTORC1 inhibition significantly reduced frequencies of Sen-β-Gal-positive cells in various models of stress-induced and replicative senescence (Fig EV5E) without rescuing the cell cycle arrest (Fig EV5F) and (ii) mTORC1 inhibition suppressed senescence-associated secretion of several pro-inflammatory cytokines (Fig EV5G). Further supporting the hypothesis that ATM and mTORC1 are downstream effectors of the DDR regulating senescence, both mTORC1 and ATM inhibition resulted in decreased mRNA expression of the SASP factor *IL6* (Fig EV5H) and mTORC1

**Figure 3. PGC-1β-dependent mitochondrial biogenesis downstream of the DDR modulates cellular senescence.**

A   Representative images of colony assays of wild-type and *PGC-1β⁻/⁻* MEFs grown at 3 or 21% O₂ (10 days after seeding 5,000 cells per well). Data are representative of 3 independent experiments.

B   Effect of 3 or 21% O₂ and X-ray irradiation (at 3% O₂) on the percentage of Ki67 (at day 6) and Sen-β-Gal-positive cells (at day 10) and the number (N) of 53BP1 foci (at day 6) in wild-type and *PGC-1β⁻/⁻* MEFs. Data are mean ± SEM of *n* = 3 independent experiments; asterisks denote a statistical significance at *P* < 0.05 using one-way ANOVA.

C   Representative images of Sen-β-Gal activity (blue cytoplasmatic staining), Ki-67 and 53BP1 foci in proliferating and senescent wild-type and *PGC-1β⁻/⁻* MEFs (scale bar = 10 μm).

D   mRNA expression of *PGC-1β*, *CXCL-1* and *p16* in proliferating and senescent (10 days after 10-Gy X-ray) wild-type and *PGC-1β⁻/⁻* MEFs. Data are mean ± SEM of *n* = 3 independent experiments; asterisks denote a statistical significance at *P* < 0.05 using one-way ANOVA.

E   Effects of overexpression of PGC-1β on percentage of Ki67- and Sen-β-Gal-positive cells, number (N) of 53BP1 foci and percentage change in mitochondrial mass (NAO intensity) in proliferating and senescent (2 days after 10-Gy X-ray) MEFs cultured at 3% O₂. Data are mean ± SEM of *n* = 3 independent experiments. Asterisks denote a statistical significance at *P* < 0.05 using one-way ANOVA or two-tailed *t*-test.

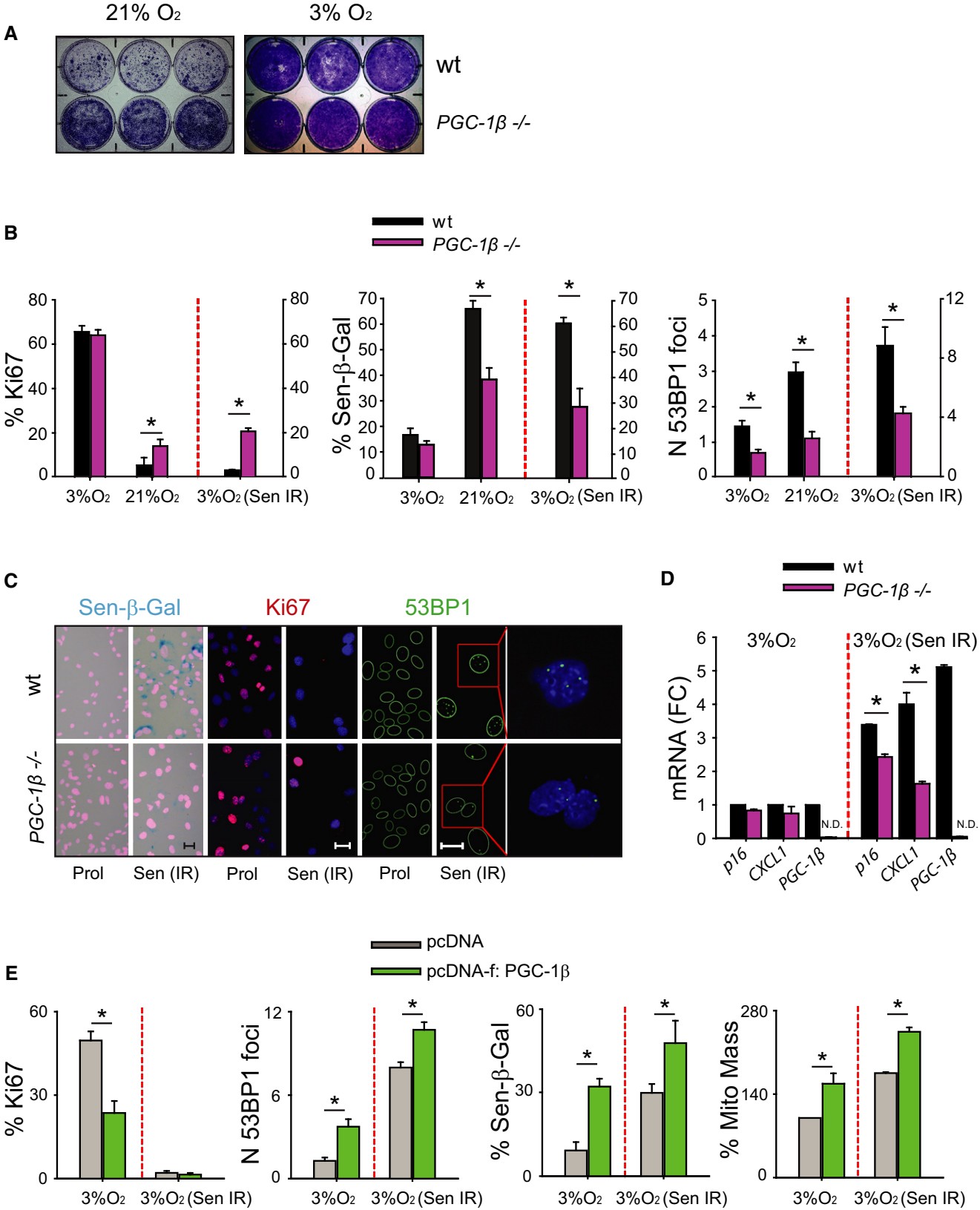

Figure 3.

▶

**Figure 4.  mTORC1 integrates DDR signals towards mitochondrial biogenesis during cellular senescence.**

A  Representative Western blot of mTORC1 activity measured by phosphorylated p70S6K (T389) from 6 to 72 h after 20 Gy in MRC5 fibroblasts. Data are representative of three independent experiments.

B  Representative Western blots of the mitochondrial proteins TOMM20, NDUFB8 (complex I), SDHA (complex II), UQCR2 (complex III) and MT-CO1 (complex IV) following 20-Gy irradiation with or without 100 nM rapamycin treatment in MRC5 fibroblasts. Data are representative of 4 independent experiments.

C  Effect of 100 nM rapamycin on mitochondrial mass (measured by NAO fluorescence) 2–4 days following replication exhaustion (RS), genotoxic stress (generated by X-ray irradiation, etoposide, neocarzinostatin (NCS), $H_2O_2$) or telomere dysfunction (TRF2$^{\Delta B\Delta M}$) in a variety of cell lines. Data are mean from 3 independent experiments per cell line or treatment.

D  (top) Representative transmission electron microscopy (T.E.M.) micrographs of proliferating and senescent (3 days after 20-Gy X-ray) MRC5 fibroblasts treated with or without 100 nM rapamycin. Mitochondria are labelled in pink. Scale bar = 2 μm; (bottom left and middle) Quantification of mitochondrial volume fraction (%V$_v$) and mitochondrial number per cross-section in proliferating and senescent (3 days after 20-Gy X-ray) MRC5 fibroblasts treated with or without 100 nM rapamycin. T.E.M. mitochondrial analysis is mean ± SEM of 24 electron micrographs per condition; (bottom right) mtDNA copy number analysis by qPCR in proliferating and senescent (3 days after 20-Gy X-ray) MRC5 fibroblasts treated with or without 100 nM rapamycin. Data are mean ± SEM of $n$ = 3 independent experiments; asterisks denote a statistical significance at $P < 0.05$ using one-way ANOVA.

E  Representative Western blots showing the expression of phosphorylated p70S6K (T389) and AKT (S473), the mitochondrial protein NDUFB8 and the DDR downstream target p21 in MRC5 fibroblasts treated with or without 10 μM of the ATM inhibitor KU55933 and in fibroblasts from a patient with ataxia telangiectasis (AT) at different time points after 20-Gy X-ray. Data are representative of three independent experiments (ATM inhibitor) and 1 experiment (AT patient).

F  Western blots showing the effect of the ATM inhibitor KU55933 on γH2A.X, AKT phosphorylation and p21 expression in MRC5 fibroblasts after 20-Gy X-ray. Data are from one experiment.

G  Effect of rapamycin and/or ATM inhibitor (KU55933) on mitochondrial mass (NAO intensity) in proliferating and senescent (3 days after 20-Gy X-ray) MRC5 fibroblasts. Data are mean ± SEM of $n$ = 3 independent experiments. Asterisks denote a statistical significance at $P < 0.05$ using one-way ANOVA.

and ATM inhibition had non-additive effects on DDF, induction of Sen-β-Gal and p21 expression (Fig 5C).

To investigate the role of mitochondria biogenesis on ROS-dependent activation of a DDR, we used *PGC-1β$^{-/-}$* MEFs. We hypothesized that PGC-1β-mediated mitochondrial biogenesis increases oxidative stress, thereby activating senescence-associated pathways. Consistently, we found that exposure to irradiation increased ROS generation in wild-type MEFs and this was markedly reduced in *PGC-1β$^{-/-}$* MEFs. In contrast, overexpression of PGC-1β resulted in increased ROS (Fig 5D) and treatment with the antioxidant Trolox prevented PGC-1β-induced DNA damage measured by frequencies of 53BP1 foci (Fig 5E), suggesting that the mitochondrial biogenesis inducer PGC-1β potentiates oxidative stress and the senescent phenotype. To investigate the role of PGC-1β in mTORC1 signalling downstream of a DDR, we increased or decreased mTORC1 activity in wild-type and *PGC-1β$^{-/-}$* MEFs by the overexpression of activated Rheb (Fig EV5I) or rapamycin supplementation, respectively. Following the activation of a DDR, the Rheb-dependent increase in the number of DDF was suppressed in *PGC-1β$^{-/-}$* MEFs and rapamycin treatment was unable to further decrease DDF in *PGC-1β$^{-/-}$* MEFs (Fig 5F), suggesting that mTORC1 and PGC-1β are part of the same pathway leading to mitochondrial ROS-dependent maintenance of a DDR. Consistent with our data that mTORC1-PGC1β-mediated mitochondrial biogenesis promotes senescence, mTORC1 inhibition by rapamycin decreased Sen-β-Gal activity in wild-type MEFs, but had no further effect in *PGC-1β$^{-/-}$* senescent MEFs (Fig EV5J).

Together, these observations suggest that maintenance of the DDR via mTORC1-PGC-1β-driven mitochondrial biogenesis impacts on the development of cellular senescence and the SASP.

**mTORC1-PGC-1β regulates mitochondrial content and contributes to senescence *in vivo***

Following our *in vitro* observations showing a DDR-dependent mitochondrial mass increase during senescence, we first asked whether mitochondrial abundance would correlate with the activation of a DDR *in vivo*. To address that, we performed dual immunofluorescence against the mitochondrial protein MT-CO1 and the DDR protein γH2A.X in liver tissue from 12-month-old mice. We observed that hepatocytes containing more MT-CO1 intensity had in general higher number of γH2A.X foci (Fig 6A). Secondly, we investigated whether the DDR-dependent activation of the mTORC1-PGC-1β pathway and increased mitochondrial content occur with age *in vivo*. Similar to our observations *in vitro*, we found an age-dependent increase in PGC-1β expression and OXPHOS components associated with increased mTORC1 activity (measured by p-S6/S6 ratio) and p21 expression in wild-type mice (Fig 6B and Appendix Fig S3A). Similar to fibroblast senescence, hepatocyte senescence seems to be associated with increased mitochondrial mass, mTORC1 activity and increased DDR signalling.

To investigate the impact of mTORC1 inhibition on mitochondrial mass *in vivo*, mice were fed with rapamycin using the same conditions as Harrison *et al* (2009) and sacrificed at different ages (Appendix Fig S3B). Rapamycin-supplemented animals presented reduced PGC-1β protein expression (Fig 6C), lowered mitochondrial volume fraction and mitochondrial numbers per cross-section (analysed by T.E.M.) (Fig 6D) and decreased mtDNA copy number (Fig 6E) when compared to controls. We also investigated whether mTORC1 inhibition impacted on mitochondrial function by potentially mediating the clearance of dysfunctional mitochondria. Mitochondrial function analysis showed no significant changes between control and rapamycin-supplemented animals. State III (ADP-stimulated), state IV and respiration uncoupled from ATP synthesis (using the uncoupler FCCP) remained unchanged using pyruvate/malate as substrate (Fig 6F). These results indicate that despite lower mitochondrial content in hepatocytes, mitochondrial function is not significantly altered by mTORC1 inhibition. Recently, it has been suggested that rapamycin may reduce ROS generation via enhanced expression of the antioxidant enzyme MnSOD (Iglesias-Bartolome *et al*, 2012), however, we failed to find any changes in MnSOD expression on rapamycin-fed mice liver tissues (Appendix Fig S3C).

We next questioned whether mTORC1 inhibition could reduce senescence markers, similar to what we had observed *in vitro*. We had previously reported that telomere-associated foci (TAF), one of

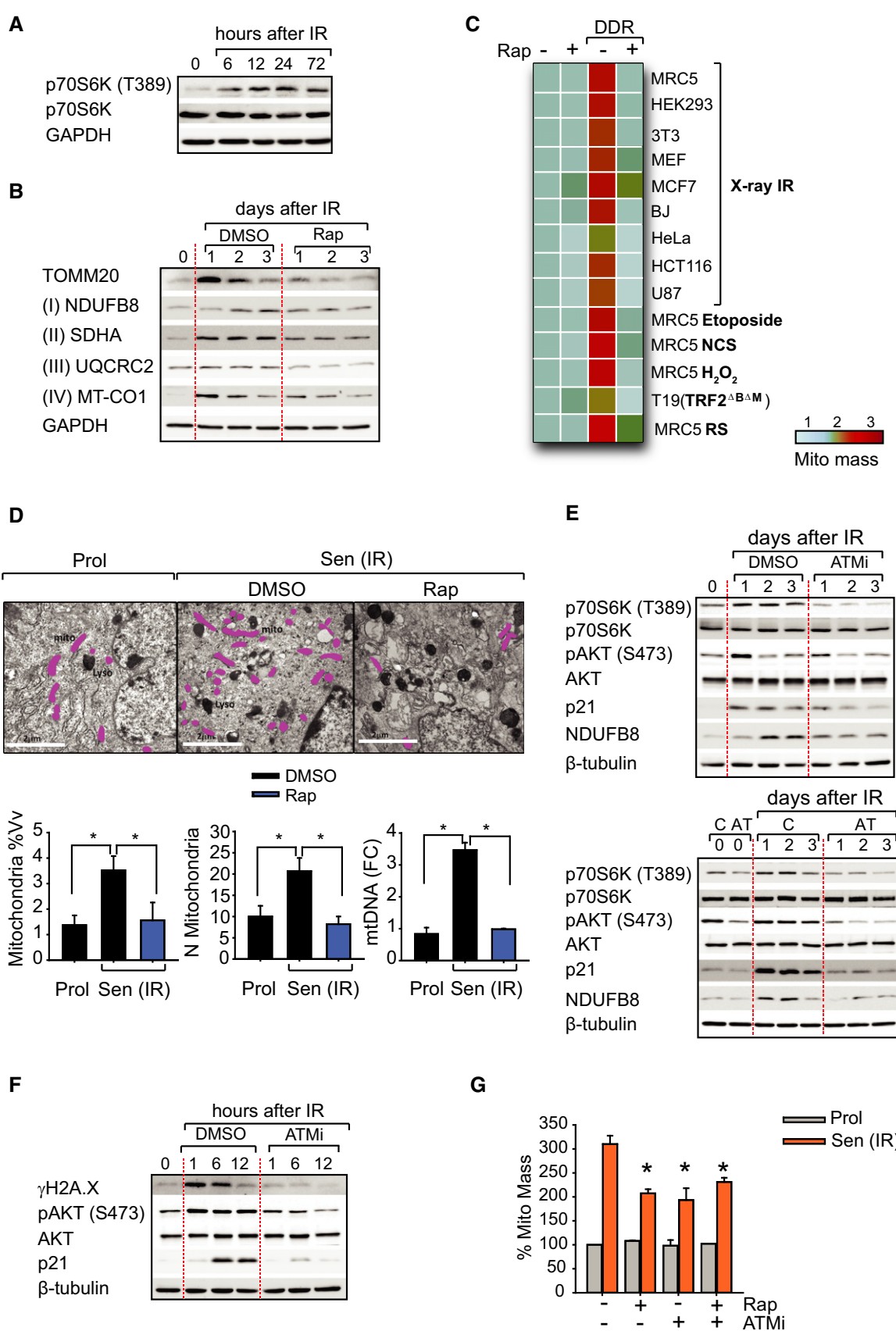

**Figure 4.**

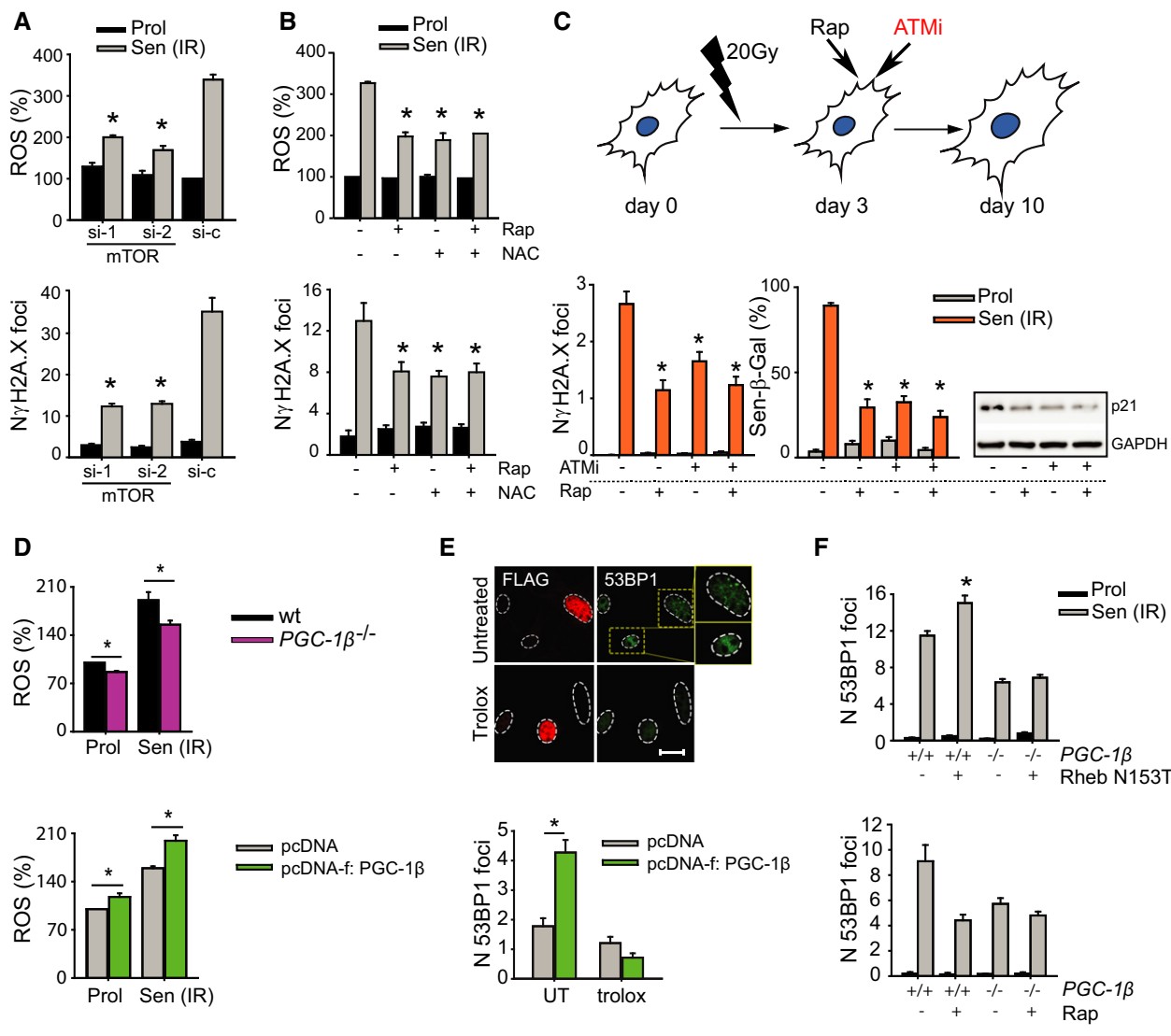

**Figure 5.  mTORC1 activation promotes ROS-dependent DDR and contributes to the senescent phenotype via PGC1-β dependent mitochondrial biogenesis.**

A ROS levels (DHE intensity) and mean number (N) of γH2A.X foci after mTOR knockdown (72 h) in proliferating and senescent (2 days after 20-Gy X-ray) MRC5 fibroblasts. Data are mean ± SEM of *n* = 3 independent experiments; asterisks denote a statistical significance at *P* < 0.05 using one-way ANOVA.

B ROS levels (DHE intensity) and mean number (N) of γH2A.X foci in proliferating and senescent (3 days after 20-Gy X-ray) MRC5 fibroblasts treated with or without 100 nM rapamycin and/or 2.5 mM of the antioxidant NAC. Data are mean ± SEM of *n* = 3 independent experiments; asterisks denote a statistical significance at *P* < 0.05 using one-way ANOVA.

C (top) Scheme illustrating the experimental design: human MRC5 fibroblasts were irradiated with 20-Gy X-ray and treated at day 3 after IR with 10 μM of the ATM inhibitor (ATMi) KU55933 and/or 100 nM rapamycin (Rap); (bottom) Effect of single or combined inhibition of ATM and mTORC1 on the mean number (N) of γH2A.X foci, Sen-β-Gal activity and p21 expression in proliferating and senescent (10 days after 20-Gy X-ray) MRC5 fibroblasts. Data are mean ± SEM of *n* = 3 independent experiments. Western blots are representative of three independent experiments; asterisks denote a statistical significance at *P* < 0.05 using one-way ANOVA.

D ROS levels (DHE intensity) in proliferating and senescent (3 days after 10-Gy X-ray) wild-type and *PGC-1β⁻/⁻* MEFs (top) and PGC-1β-overexpressing MEFs (bottom). Data are mean ± SEM of *n* = 3 independent experiments; asterisks denote a statistical significance at *P* < 0.05 using one-way ANOVA.

E (top) Representative images and quantification of immunofluorescence staining against FLAG-PGC-1β (red) and 53BP1 (green) in PGC-1β-overexpressing MEFs cultured at 3% O₂, with or without 250 μM of the antioxidant Trolox (scale bar = 10 μm). Data are mean ± SEM, *n* = 3 independent experiments; asterisks denote a statistical significance at *P* < 0.05 using one-way ANOVA.

F (top) Effect of overexpression of mutated Rheb (N153T) and (bottom) effect of 100 nM rapamycin on the mean number (N) of 53BP1 foci in proliferating and senescent (3 days after 10-Gy X-ray) wild-type and *PGC-1β⁻/⁻* MEFs. Data are mean ± SEM of *n* = 3 independent experiments (at least 125 cells were analysed per condition). Asterisks denote a statistical significance at *P* < 0.05 using one-way ANOVA.

the important effectors of cellular senescence, increase with age in wild-type mice tissues and in models of accelerated ageing (Hewitt *et al*, 2012; Jurk *et al*, 2014). In order to test the impact of mTORC1 inhibition on telomere-induced senescence *in vivo*, we analysed

liver tissue from rapamycin-fed mice and showed that mTORC1 inhibition was able to prevent age-dependent increase in TAF (Fig 6G). Interestingly, 4 months of rapamycin-supplemented diet was as effective in reducing TAF as 12 months (Fig 6G). Moreover,

Figure 6.  mTORC1-PGC-1β regulates mitochondrial content and contributes to senescence *in vivo*.

A　(top) Representative image of immunofluorescence double staining for the mitochondrial protein MT-CO1 and the DDR marker γH2A.X (scale bar = 5 μm) and (bottom) quantification of MT-CO1 intensity versus the number of γH2A.X foci in hepatocytes from 12-month-old mice. Data are mean ± SEM of $n = 3$ mice (at least 30 cells were analysed per mouse); asterisks denote a statistical significance at $P < 0.05$ using one-way ANOVA.

B　Representative Westerns blots of pS6, S6, p21, PGC-1β, MT-CO1 and NDUFB8 protein expression in mouse liver tissue at 3 and 12 months of age. Data are from $n = 3$ mice per group.

C　Effect of 4 months of rapamycin-supplemented diet on PGC-1β expression in the liver tissue of 16-month-old mice. Data are from $n = 3$ mice per group.

D　(top) Quantification of mitochondrial number per cross-section and mitochondrial volume fraction (%$V_v$) in hepatocytes from 16-month-old mice with or without 4 months of rapamycin diet. Data are mean ± SEM of $n = 3$ mice per group (at least 20 cells were analysed per mouse); (bottom) Representative electron micrographs of hepatocytes from 16-month-old mice with or without 4 months of rapamycin diet (mitochondria are labelled in pink). Scale bar = 5 μm. Asterisk denotes a statistical significance at $P < 0.05$ using two-tailed $t$-test.

E　mtDNA copy number (measured by qPCR) in mice liver tissue at 3, 12, 16 months and at 16 months after 4 months of rapamycin diet. Data are mean ± SEM of $n = 3-4$ mice per group; Asterisks denote a statistical significance at $P < 0.05$ using one-way ANOVA.

F　Oxygen consumption rates (OCR) in isolated liver mitochondria from 16-month-old mice fed with or without rapamycin for 4 months, in the presence of pyruvate/malate. State III was induced by the injection of ADP. State IV was induced by the inhibition of the ATP synthase with oligomycin, and uncoupled respiration rates were determined by the injection of FCCP. Antimycin A (AA) was used to determine background, non-mitochondrial OXPHOS, OCR. Data are mean ± SEM of $n = 5$ mice per group.

G　(top) Representative immuno-TeloFISH images of hepatocytes from 3- and 15-month-old mice with or without rapamycin (12-month diet). Co-localizing foci are amplified in the right panel (amplified images are from single Z-planes where co-localization was found); (bottom) Dot plot graph of telomere-associated foci (TAF) in 3-, 15- and 16-month-old mice (15- and 16-month-old mice were fed with rapamycin for 12 and 4 months, respectively). Data are from $n = 3$ to 9 mice per group (at least 50 cells were analysed per mice). Values are the mean for individual animals, with the horizontal line representing group mean. Asterisk denotes $P < 0.05$ using an independent samples $t$-test.

H　(top) 4-month-old and 15-month-old mice livers [control (−) or rapamycin (+)] stained with Sen-β-Gal solution (Sen-β-Gal activity is evidenced by blue staining). Data are from $n = 3$ mice per group; (bottom) Representative image showing Sen-β-Gal staining (Sen-β-Gal activity is evidenced by blue staining) in hepatocytes and corresponding immuno-TeloFISH (arrows represent co-localizing foci); asterisks denote a statistical significance at $P < 0.05$ using one-way ANOVA.

I　Representative Western blot showing the effect of 4-month rapamycin feeding on p21 expression in 16-month-old mice. Data are from $n = 3$ mice per group.

J　Effect of 4-month rapamycin feeding on mRNA expression of the SASP components *CXCL1*, *CXCL5* and *inhibin A* in liver tissue of 16-month-old mice. Data are from $n = 5$ mice per condition; asterisks denote a statistical significance at $P < 0.05$ using two-tailed $t$-test.

K　Mean number of TAF in hepatocytes of wild-type and $PGC\text{-}1\beta^{-/-}$ mice with 18 months of age. Data are mean ± SEM of $n = 4$ mice per group; asterisks denote a statistical significance at $P < 0.05$ using two-tailed $t$-test.

L　Scheme represents overall hypothesis: feedback loop between DDR, mTORC1 and mitochondrial biogenesis stabilizes cellular senescence, which are key factors for the development of the senescence-associated pro-oxidant and pro-inflammatory phenotypes.

TAF could not be attributed to changes in telomere length or telomerase activity (Appendix Fig S3D and E). Consistent with a role for mTORC1 in senescence *in vivo*, we observed (i) decreased Sen-β-Gal activity, with Sen-β-Gal-positive hepatocytes being generally positive for TAF (Fig 6H), (ii) reduced expression of the cyclin-dependent kinase inhibitor p21 (Fig 6I) and (iii) diminished SASP factors expression (Fig 6J) in the liver tissue of rapamycin-fed animals.

In order to test *in vivo* whether the expression of PGC-1β would have an impact on senescence, we analysed the liver tissue from aged $PGC\text{-}1\beta^{-/-}$ mice. $PGC\text{-}1\beta^{-/-}$ mice showed reduced mtDNA copy number, mitochondrial proteins in liver tissue and lower energy expenditure than wild-type littermates (Appendix Fig S3F). Consistent with our hypothesis that mitochondrial content impacts on DDR and our data revealing a role for PGC-1β in senescence *in vitro*, we found reduced senescent markers such as TAF (Fig 6K) in $PGC\text{-}1\beta^{-/-}$ mice. The absence of PGC-1β also ameliorated age-dependent decline in glucose and insulin tolerance (Appendix Fig S3G). Similar to rapamycin-fed mice, we did not find any changes in MnSOD expression in $PGC\text{-}1\beta^{-/-}$ mice liver tissues (Appendix Fig S3H). Altogether, our data strongly support a causal link between the DDR, mTORC1 and mitochondrial mass increase in the development of senescence *in vitro* and *in vivo* (Fig 6L).

## Discussion

Mitochondria have been widely associated with cellular senescence and ageing. While pharmacological interventions reducing ROS generation have been proven to be beneficial, it is unclear whether

mitochondria are required for senescence. ROS derived from non-mitochondrial sources (Takahashi *et al*, 2006) and imbalances in antioxidant defence (Blander *et al*, 2003) have equally been implicated in the process. Chemical and genetic interventions impacting on the mitochondrial electron transport chain have been shown to enhance OIS (Moiseeva *et al*, 2009); however, to date, no study has effectively evaluated the necessity of mitochondria for the induction of the senescent phenotype. In order to investigate the role of mitochondria in senescence, we used a similar approach to recent explorations of the mechanisms underlying cell death (Tait *et al*, 2013). By targeted depletion of mitochondria, we demonstrate that these organelles are key factors for the development of major ageing-promoting characteristics of the senescent phenotype, including the pro-oxidant and pro-inflammatory phenotypes, but less so for the proliferation arrest despite increased glycolysis and ATP generation. It is still unclear what is the mechanism contributing to the maintenance of the proliferation arrest. We speculate that a multitude of factors including the production of specific growth-stimulating mitochondrial metabolites or interactions between mitochondria and cytoplasmic elements required for cell division may underlie the process (Mitra *et al*, 2009). The interplay between mitochondria and the mTOR signalling pathway may also be involved in this process, since we have shown that mitochondria clearance reduces mTOR activity and inhibition of mTOR activity inhibits proliferation contributing to quiescence. Recent studies have also shown that the mechanisms by which mTOR impacts on the cell cycle are uncoupled from its effect on the SASP (Herranz *et al*, 2015; Laberge *et al*, 2015). While we acknowledge that mitochondrial ablation is a drastic intervention with possible widespread consequences for the cells,

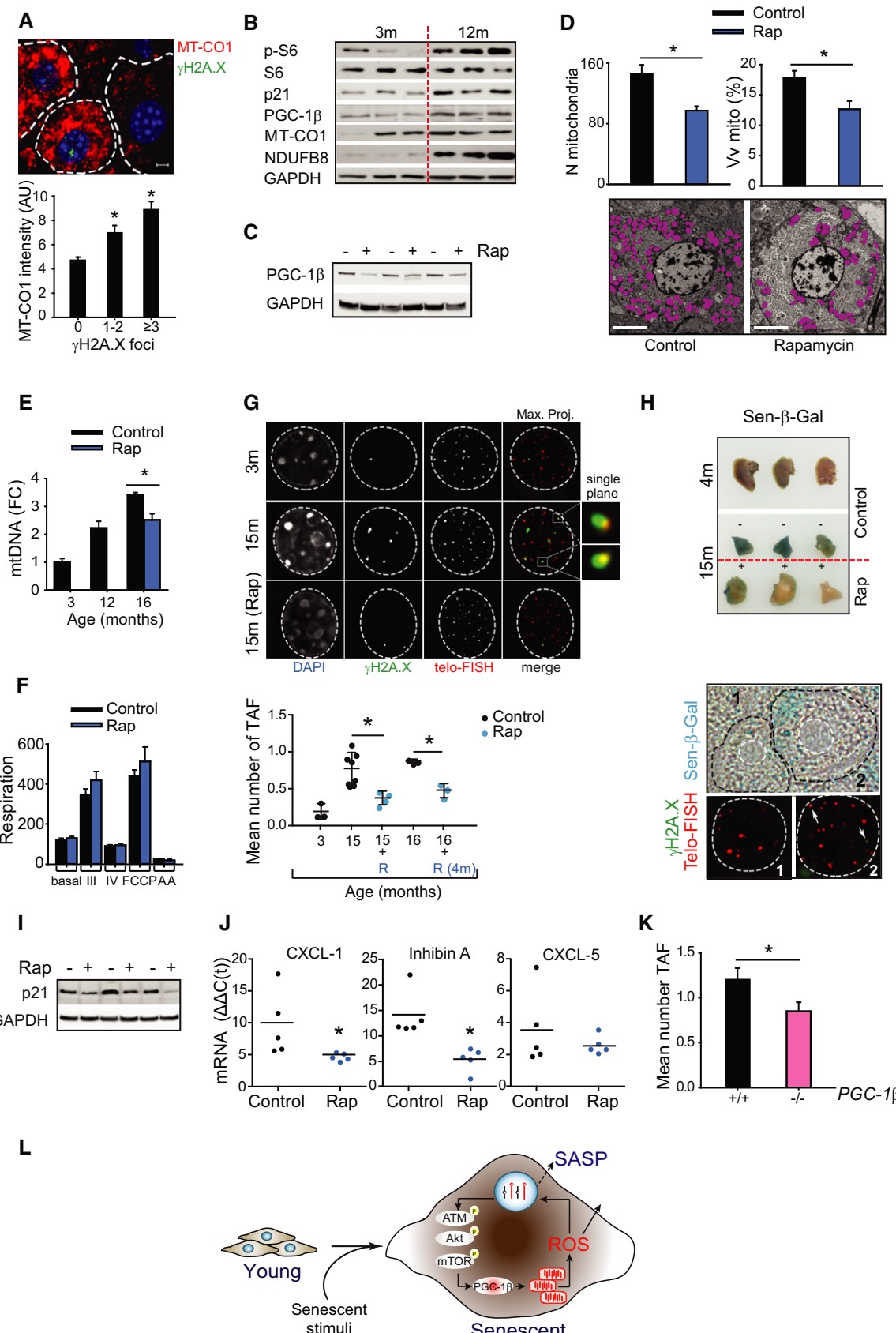

**Figure 6.**

our results demonstrate that mitochondria play a central role in many of the transcriptional changes observed during senescence, particularly the SASP, and have, as such, a great therapeutic potential.

Furthermore, we have identified a novel pathway involving ATM, Akt, mTORC1 and PGC-1β in the induction of senescence. Although we do not exclude the existence of alternative pathways impacting on mitochondrial content during senescence, such as impaired mitophagy, our results establish a novel link between the activation of a DDR and mitochondrial biogenesis involving the mTORC1 signalling cascade. While mTORC1 has previously been implicated in senescence (Pospelova *et al*, 2012), the mechanisms regulating its activation as well as its links to the pro-oxidant and pro-inflammatory features of the phenotype are still unclear. Two recent studies have shown that mTOR inhibition suppressed the senescence-associated secretion of inflammatory cytokines by suppressing the translation of SASP regulatory proteins IL-1A and MK2 (Herranz *et al*, 2015; Laberge *et al*, 2015). On the other hand, our data show that interventions targeting the mTOR pathway affect the maintenance of a DDR via the generation of ROS, which has been shown to be necessary for the induction of senescence and the SASP (Rodier *et al*, 2009; Passos *et al*, 2010). Our findings are consistent with a model by which mTORC1 signalling impacts on mitochondria content and by doing so, affects the persistence of a DDR and the development of the SASP. We cannot disregard, however, direct modes of interaction between mTORC1 and targets of the DDR such as p53, independently of mitochondria (Lee *et al*, 2007; Lai *et al*, 2010). While these studies report dynamic interactions between the mTOR and p53 pathways very shortly after a DDR, they do not explain how the cell cycle arrest is maintained and contributes to the development of senescence, which is a lengthier process usually taking between 7 and 10 days post-DDR activation (Coppé *et al*, 2008). While both our data and others indicate the links between DDR activation and the SASP (Rodier *et al*, 2009; Osorio *et al*, 2012), it is possible that alternative mechanisms play a role in the process, given reported direct interactions between mTOR signalling, mitochondria and major SASP regulators such as the NF-κB pathway (Correia-Melo *et al*, 2014).

Our *in vivo* data are consistent with reports in haematopoietic stem cells in which mTORC1 activity increases mitochondrial biogenesis, contributing to ROS-dependent decreased stemness and haematopoiesis (Chen *et al*, 2008), and that mTOR inhibition can alleviate mitochondrial disease (Johnson *et al*, 2013). Furthermore, the observation of a key role for mTORC1 *in vivo* in the maintenance of a DDR is supported by reports that calorie restriction (CR), which impacts on mTORC1 activity, reduces DDR-positive cells *in vivo* in several tissues (Wang *et al*, 2010; Jurk *et al*, 2012) as well as ROS (Lanza *et al*, 2012). Our observations of reduced mitochondrial content in hepatocytes following mTORC1 inhibition are in accordance with various reports of reduced protein synthesis and mitochondrial transcriptional regulation observed in mice with impaired mTORC1 activity (Cunningham *et al*, 2007; Romanino *et al*, 2011). Moreover, it has been recently shown by whole-genome expression profiling and large-scale proteomic analysis that mice under CR have less mitochondrial protein synthesis (Lanza *et al*, 2012). This latter study, together with other more recent reports (Hancock *et al*, 2011; Lanza *et al*, 2012; Price *et al*, 2012), has questioned early associations between CR and

enhanced mitochondrial content (Nisoli *et al*, 2005). Furthermore, despite evidence suggesting that increased mitochondrial abundance can be an advantageous adaptive response to energy deficit, genetically induced mitochondrial biogenesis has been associated with age-related diseases such as cardiomyopathy (Lehman *et al*, 2000), renal fibrosis (Hickey *et al*, 2011) and diabetes (Sawada *et al*, 2014), all of which have been associated with cellular senescence (Chimenti *et al*, 2003; Sone & Kagawa, 2005; Ning *et al*, 2013).

Age-dependent mitochondrial degeneration has been associated with impaired clearance of dysfunctional mitochondria, a process known to be negatively regulated by mTORC1. Nonetheless, our data show that rapamycin leads to no significant changes in the function of isolated mitochondria from liver, which has been independently confirmed (Johnson *et al*, 2013; Miwa *et al*, 2014), despite reducing DNA damage, senescence markers and ROS generation. Our data suggest that mTORC1-dependent enhanced mitochondrial content may be impacting on ageing-promoting features of senescence *in vivo* without necessarily affecting mitochondrial function; however, further detailed studies on the role of mTOR signalling on mitochondrial function and the mechanisms underlying ROS generation are clearly warranted.

There is great interest in therapeutically combating the pro-oxidant and pro-inflammatory effects of cellular senescence, given their suggested role as drivers of age-related diseases (Tchkonia *et al*, 2013; Correia-Melo *et al*, 2014). Our observations demonstrating that mitochondria are required for these features of senescence have very important implications for our understanding of the origins and mechanisms underlying the senescence phenotype and suggest mitochondria as major putative therapeutic targets for interventions impacting on the senescent phenotype, while maintaining its tumour suppressor capability.

# Materials and Methods

### Cell culture

Human embryonic lung MRC5 fibroblasts, human embryonic kidney HEK293, human breast cancer MCF7, human primary glioblastoma U87, human colon carcinoma HCT116, human primary BJ fibroblasts, Phoenix amphotropic cells, AT patient fibroblasts, T19 and HeLa cells were grown in Dulbecco's modified Eagle's medium (DMEM, Sigma, Dorset, UK) supplemented with 10% heat-inactivated foetal calf serum (FCS), 100 units ml$^{-1}$ penicillin, 100 μg ml$^{-1}$ streptomycin and 2 mM glutamine at 37°C in a humidified atmosphere with 5% $CO_2$. Human 143B parental and rho0 osteosarcoma cells were grown in Dulbecco's modified Eagle's medium (DMEM, Sigma, Dorset, UK) supplemented with 10% heat-inactivated foetal calf serum (FCS), 100 units ml$^{-1}$ penicillin, 100 μg ml$^{-1}$ streptomycin, 2 mM glutamine, 50 μg ml$^{-1}$ uridine and 0.11 mg ml$^{-1}$ sodium pyruvate at 37°C in a humidified atmosphere with 5% $CO_2$. T19 cells containing a doxycycline-inducible TRF2$^{\Delta B\Delta M}$ were a kind gift from T. de Lange, Rockefeller University, NY, USA (van Steensel *et al*, 1998). AT patient fibroblasts were a kind gift from Dr. Lisa Woodbine, University of Sussex, UK.

Mouse embryonic fibroblasts (MEFs) were obtained from C57/BL6 wild-type mice. MEFs from *PGC-1β*$^{-/-}$ and matched wild-type

mice were cultured in advanced DMEM/F-12 (Invitrogen) plus 10% FSC, 100 units ml$^{-1}$ penicillin, 100 μg ml$^{-1}$ streptomycin and 2 mM glutamine at 37°C in a humidified atmosphere with 5% $CO_2$ and 3% oxygen in a 3-gas cell culture incubator (Zapf Instruments, Sarstedt, Germany). Cells were regularly checked for mycoplasma contamination.

### Induction of senescence

Stress-induced senescence was induced by X-ray irradiation with 10 or 20 Gy (depending on cell line), neocarzinostatin (80 ng ml$^{-1}$) for 1 h and $H_2O_2$ (400 μM) in serum-free media for 1 h. Following treatment, culture medium was refreshed. Etoposide treatment (50 μM) was performed continuously every 3 days for 10 days. Oncogene-induced senescence: human foetal lung IMR90 diploid fibroblasts were cultured in DMEM (phenol red free) supplemented with 20% FBS, 100 units ml$^{-1}$ penicillin, 100 μg ml$^{-1}$ streptomycin and 2 mM glutamine at 37°C in a humidified atmosphere with 5% $CO_2$ and 3% oxygen in a 3-gas cell culture incubator (Zapf Instruments, Sarstedt, Germany). For ER-RAS induction, ER-RAS-IMR90 cells were supplemented with 100 nM of 4-hydroxytamoxifen (4-OHT) and maintained in 4-OHT-containing medium until harvesting. Replicative senescence was achieved through replication exhaustion and confirmed by cells being positive for Sen-β-Gal, negative for BrdU and Ki67 and performing less than 0.5 population doublings for at least 4 weeks.

### Plasmid transfection and viral infection

For the overexpression of activated Rheb and PGC-1β, MEFs were transfected with an empty vector and pcDNA3-flag-Rheb-N153T (Addgene #19997) or pcDNAf:PGC-1β (Addgene #1031) respectively, using Lipofectamine™ 2000 (Invitrogen, 11668-019) following the manufacturer's instructions. YFP-Parkin retroviral production: phoenix amphotropic cells were incubated for 24 h in antibiotic-free medium before proceeding to transfection with a LZRS or LZRS-YFP_Parkin vector using Lipofectamine™ 2000 (Invitrogen, 11668-019) following the manufacturer instructions. Cells were transduced with YFP-Parkin retroviral and shPGC-1β lentiviral particles (Santa Cruz Biotechnology #sc-62783-V) as described in the ViraPower Lentiviral Expression systems user manual (Invitrogen). All experiments involving virus were performed following class II safety procedures. Transduction efficiency was confirmed by fluorescence microscopy, Western blotting and/or qPCR.

### RNA sequencing

Samples were sequenced on an Illumina Next Seq. Paired-end reads were aligned to the human genome (hg19) using the splicing-aware aligner Tophat2 (Kim *et al*, 2013). Duplicate reads were identified using the picard tools (1.98) script mark duplicates (https://github.com/broadinstitute/picard) and only non-duplicate reads were retained. Reference splice junctions are provided by a reference transcriptome (Ensembl build 73). FPKM values were calculated using Cufflinks (Trapnell *et al*, 2013). Differential genes expression was called using the cuffdiff (Trapnell *et al*, 2013) maximum-likelihood estimate function. Genes of significantly changing expression were defined as FDR-corrected *P*-value < 0.05 and at least twofold change.

Only ensembl 73 genes of status "known" and biotype "coding" were used for downstream analysis.

Reads were trimmed using Trim Galore (v0.3.0) (http://www.bioinformatics.babraham.ac.uk/projects/trim_galore/) and quality assessed using FastQC (v0.10.0) (http://www.bioinformatics.bbsrc.ac.uk/projects/fastqc/).

*Venn diagrams*
Venn diagrams and associated empirical *P*-values were generated using the USeq (v7.1.2) tool IntersectLists (Nix *et al*, 2008). The *t*-value used was 22,008, as the total number of genes of status "known" and biotype "coding" in ensembl genes 73. The number of iterations used was 1,000.

*RNA-seq heatmaps*
For each gene, the FPKM value was calculated based on aligned reads, using Cufflinks (Trapnell *et al*, 2013). Z-scores were generated from FPKMs. Hierarchical clustering was performed using the R library heatmap.2 and the distfun="pearson" and hclustfun="average".

*Principal component analysis*
Principal component analysis (PCA) was performed using the FPKM values of all ensembl 73 genes of status "known" and biotype "coding".

The RNA-seq data from this publication have been submitted to the GEO database (http://www.ncbi.nlm.nih.gov/geo/) and assigned the identifier GSE76125.

### Treatment with pathway inhibitors

For mTORC1 inhibition, cells were treated with 100 nM rapamycin (Sigma, R8781) immediately after the senescence stimuli (X-ray, neocarzinostatin, $H_2O_2$ and etoposide). Cells were collected at different time points after treatments. Replicative senescent MRC5 fibroblasts were treated with 100 nM rapamycin for 10 days before being harvested for analysis. Rapamycin-supplemented media were always refreshed 24 h prior to cell collection to avoid starvation confounding effects on mTORC1 activity.

For ATM inhibition, MRC5 fibroblasts irradiated with 20-Gy X-ray were treated with 10 μM of the ATM chemical inhibitor KU55933 (R&D Systems; 3544). Cells were collected at different time points after treatment.

Antioxidant treatments were performed in cells irradiated with 20-Gy X-ray treated with 2.5 mM NAC (Sigma; A7250) or 400 μM PBN (Sigma, 180270) immediately after irradiation. The cells were collected at different times after treatment.

### Parkin-mediated mitochondria clearance

Proliferating stably expressing YFP-Parkin human MRC5 fibroblasts were treated with 12.5 μM CCCP (Sigma; C2759) for 48 h (refreshed every 24 h). Stably expressing YFP-Parkin human MRC5 fibroblasts were irradiated with 20-Gy X-ray and treated 2 days after with 12.5 μM CCCP for 48 h (refreshed every 24 h). Stably expressing YFP-Parkin human MRC5 fibroblasts were treated with 400 μM of $H_2O_2$ for 1 h in serum-free medium and treated 2 days after with 12.5 μM CCCP for 48 h (refreshed every 24 h). Replicative senescent

YFP-Parkin-MRC5 fibroblasts were treated with 12.5 µM CCCP for 48 h (refreshed every 24 h). Media (no CCCP) was refreshed after CCCP treatment; the cells were then collected at the indicated time points for analysis. Oncogene-induced senescence (OIS) experiments were performed by treating Parkin-expressing IMR-90/ER-RAS fibroblasts with 12.5 µM CCCP for 48 h, followed by ER-RAS induction with 100 nM of 4-hydroxytamoxifen (4-OHT). The cells were harvested at day 7 after ER-RAS induction. Mitochondria-depleted cells require to be seeded at a cellular density of 60–70%.

## Knockdown by small interfering RNA (siRNA)

MRC5 cells were transiently transfected with siRNAs using HiPer-Fect Transfection ReagentTM (Qiagen, 301707). Cells were transfected with 10 nM siRNA (negative control siRNA, Qiagen—SI03650325; Hs_FRAP1_4 FlexiTube siRNA, Qiagen—SI00070462; Hs_FRAP1_6 FlexiTube siRNA, Qiagen—SI02662009) following the HiPerFect Transfection Reagent Handbook instructions. The cells were transfected 24 h prior to 20-Gy X-radiation and harvested for analysis 72 h after transfection (2 days after IR). siRNA transfection efficiency was assessed by qPCR and/or Western blotting.

## Mitochondrial mass and ROS measurements

For mitochondrial mass assessment, cells were stained with 10 µM of nonyl acridine orange (NAO, Molecular Probes, Invitrogen) for 10 min at 37°C and measured by flow cytometry. Alternatively, the cells were loaded with MitoTracker green in serum-free medium for 30 min at 37°C and imaged on a Leica DM5500B fluorescence microscope. Transmission electron microscopy was performed using conventional techniques, and mitochondrial number and volume fraction were analysed from 15 to 20 electron micrographs per animal ($n = 3$) or human fibroblasts using ImageJ (http://rsb.info.nih.gov/ij/). mtDNA copy number was measured in human and mouse fibroblasts by qPCR (primers in Appendix Table S1). ROS were measured using 10 µM dihydroethidium (DHE), which detects cellular superoxide levels, or 5 µM MitoSOX (Molecular Probes, Invitrogen), which detects mitochondrial superoxide levels, or 10 µM dihydrorhodamine 123 (DHR123), which detects cellular peroxides. Cells were incubated for 30 or 10 min at 37°C in the dark, with DHE (5 µM) or MitoSOX (5 µM), respectively, in serum-free media and analysed by flow cytometry (30,000 cells were analysed per condition).

## Measurements of cellular bioenergetics

Cellular oxygen consumption rates (OCR) and media acidification rates (extracellular acidification rate, ECAR) were measured in parallel using a Seahorse XF24 analyzer in unbuffered basic media supplemented with 5 mM glucose, 2 mM L-glutamate and 3% FBS. The following compounds were supplemented to test mitochondrial activity and cellular bioenergetics flux: 0.5 µM oligomycin to inhibit ATP synthase (hence inhibiting mitochondrial ATP generation), 2.5 µM carbonyl cyanide p-trifluoromethoxy-phenylhydrazone (FCCP), a respiratory chain uncoupler, 80 mM 2-deoxyglucose (2DG), a glucose analogue competitively inhibiting glucose uptake and glycolytic flux, and 0.5 µM rotenone and 2.5 µM antimycin, mitochondrial complex I and complex III inhibitors, respectively.

ATP production by mitochondria was calculated by multiplying ATP turnover ((basal OCR) – (OCR with oligomycin) by the established phosphorus/oxygen (P/O) ratio of 2.3 (Brand, 2005). ATP production by glycolysis is considered to have a 1:1 ratio with lactate production. The extracellular acidification rate is mainly due to lactate and bicarbonate production and, when calibrated as the proton production rate, indicates glycolytic rate (Birket *et al*, 2011; Wu *et al*, 2007). Data were normalized to cell number.

Steady-state cellular ATP levels were performed using the ATP determination Kit (Invitrogen, A22066) following the manufacturer's instructions.

## Measurements of OCR in isolated mitochondria

Mouse liver mitochondria were isolated by the method described by Chappell and Hansford (1972) in medium comprising 0.25 M sucrose, 5 mM Tris–HCl and 2 mM EGTA (pH 7.4 at 4°C) (STE buffer). The crude mitochondria were purified by adapting the method described in Pagliarini *et al* (2008). Essentially, 0.5 ml of crude mitochondria (about 30–40 mg ml$^{-1}$) was carefully layered on top of a stepwise density gradient of 2 ml 80%, 6 ml 52% and 6 ml 26% Percoll in a 50-ml centrifuge tube. The gradient was centrifuged at 41,100 g for 45 min in a Beckman Coulter Avanti® J-E centrifuge, using a JA-20 rotor. Mitochondria were collected from the interface of the 26–52% interface, diluted to capacity in a 2-ml microcentrifuge tube with STE buffer and centrifuged at 12,000 g in a refrigerated table top centrifuge for 10 min. The supernatant was carefully discarded, and the mitochondria were washed with an additional 2 ml of STE buffer and centrifuged again. The resulting pellet was resuspended in a small volume of STE buffer for functional experiments. Mitochondrial oxygen consumption rates were measured using the Seahorse XF analyzer based on the method described in Rogers *et al* (2011), with adjustment explained in the Comments to the articles published in PLoS online (A source of data variation in mitochondrial respiration measurements). The mitochondria were energized with 5 mM pyruvate and 5 mM malate, and State III respiration was obtained by adding 4 mM ADP.

## Assessment of the SASP

Quantibody Human Cytokine Arrays for 20 cytokines (RayBiotech; QAH-CYT-1) were performed. Conditioned media were collected prior to irradiation (time 0) and 3 and 10 days after 20-Gy X-ray following 24-h serum deprivation. mRNA levels of human IL-6 and IL-8 and mouse IL-6, CXCL1, CXCL5 and inhibin A were measured by qPCR (primers in Appendix Table S2). Concentrations of IL-6 and IL-8 in cell culture media were determined using a sandwich ELISA (R&D Systems; DY206/DY208) according to the manufacturer's instructions. Limits of detection for these assays were 10 pg ml$^{-1}$.

## Quantitative real-time PCR

Total RNA was extracted using the RNeasy minikit (QIAGEN, 74106). Complementary DNAs were generated using the Omniscript RT Kit (Qiagen, 205110) as described on the Omniscript® Reverse Transcription Handbook. Quantitative real-time PCR were conducted using the Power Syber® Green PCR Master Mix (Invitrogen, 4367659) in a C1000TM Thermal Cycler, CFX96TM Real-Time

System (Bio-Rad) and Bio-Rad CXF Manager software. Expression was normalized to β-actin for mouse or GAPDH for human samples. Sequences of the primers used are included in the Appendix Table S2.

## Immunofluorescence and immuno-FISH

For immunofluorescence, cells grown on coverslips were fixed in 2% paraformaldehyde and incubated with a primary antibody for 1 h at room temperature, followed by either an Alexa Fluor® 488, 594 or 647 secondary antibody (Invitrogen) 45-min incubation. Antibodies used are given in the Appendix Table S3.

For immunofluorescence coupled with telomere-fluorescence *in situ* hybridization (immuno-FISH), cells grown on coverslips were fixed in 2% paraformaldehyde and γ-H2A.X immunofluorescence staining was performed. Following immunostaining, tissue sections were washed in PBS and fixed in 4% formaldehyde aqueous solution buffered for 20 min. The tissue sections were then dehydrated with 70, 90 and 100% ethanol solutions. Samples were denatured for 10 min at 80°C in hybridization buffer [70% deionized formamide, 25 mM $MgCl_2$, 1 M Tris pH 7.2, 5% blocking reagent (Roche, Welwyn, UK)] containing 4 ng μl$^{-1}$ Cy-3-labelled telomere-specific (CCCTAA) peptide nucleic acid probe (Panagene, F1002-5), followed by hybridization for 2 h at room temperature. The slides were washed for 10 min with 70% formamide in 2× SSC following by a wash in 2× SSC and a final wash in PBS for 10 min. Nuclei were counterstained with DAPI, and the sections were mounted and imaged in a Leica DM5500B fluorescence microscope. In-depth Z stacking was used (a minimum of 40 optical slices with 100× objective) followed by Huygens (SVI) deconvolution. Telomere-associated foci were analysed blinded by 3 analysts.

## BrdU and EdU incorporation

BrdU incorporation was performed for 24 h. Cells were fixed in cold 70% ethanol, treated with 1.5 M hydrochloric acid for 30 min and neutralized with 0.1 M borate buffer for 10 min. Cells were washed in PBS and BrdU immunostaining was performed (as described above). EdU incorporation was performed for 24 h prior to the detection with the Click-IT EdU Alexa Fluor 594 Imaging Kit (Invitrogen, C103339) following the manufacturer's instructions.

## Senescence-associated β-galactosidase staining (Sen-β-Gal)

For Sen β-Gal staining, the cells were fixed with 2% paraformaldehyde for 5 min and stained as described in Dimri *et al* (1995). For liver tissues, Sen β-Gal staining was performed as described in Baker *et al* (2011).

## Western blotting analysis

Western blotting was performed using conventional techniques. Primary and secondary antibodies are described in detail in the Appendix Table S4.

## Mice groups, treatments and housing

C57/BL6 mice were split into 2 groups (*n* = 10/group) according to age and diet: (i) 15-month-old mice: 3-month-old mice fed with control or rapamycin diet for 12 months, and (ii) 16-month-old mice: 12-month-old mice fed with control or rapamycin diet for 4 months. The different mice groups were matched for age and randomly assigned for the treatments. Control and rapamycin diets were purchased from TestDiet—control diet: 5LG6/122 PPM EUDRAGIT 3/8 #1814831 (5AS0) and encapsulated rapamycin diet: 5LG6/122 PPM ENCAP RAP 3/8 #1814830 (5ARZ). Mice were fed *ad libitum* and monitored weekly. *PGC-1β*$^{−/−}$ mice were generated and provided by Transgenic RAD, Discovery Science, Astrazeneca. Animals were fed *ad libitum* on a normal chow diet (10% of calories derived from fat; D12450B, Research Diets). Wild-type and *PGC-1β*$^{−/−}$ C57/BL6 mice were sacrificed at 18 months of age.

Animal procedures were performed in accordance with the UK Home Office regulations and the UK Animal Scientific Procedures Act [A(sp)A 1986]. Animals were housed in a temperature-controlled room with a 12-h light/dark cycle. No statistical method was used to predetermine sample size. No animals or samples were excluded from the analysis.

## Mice tissues collection and preparation

Tissues were collected during necropsy and fixed with either 4% formaldehyde aqueous solution buffered (VWR; 9713.9010) and paraffin embedded for histochemical analysis or with glutaraldehyde (Sigma, G5882) for T.E.M. (morphometric analysis). Part of the tissues were also frozen in liquid nitrogen and stored at −80°C for biochemical analysis.

## Statistical analyses

We conducted one-way ANOVA, two-tailed *t*-test, linear and nonlinear regression analysis test using Sigma Plot v11.0. Normal distribution and equal variance were assessed using Statistical software from Sigma Plot vs11.0. Wilcoxon–Mann–Whitney tests were conducted using IBM SPSS Statistics 19.

## Ethics statement

All work complied with the guiding principles for the care and use of laboratory animals. The project was approved by the Faculty of Medical Sciences Ethical Review Committee, Newcastle University. Project licence number 60/3864.

**Expanded View** for this article is available online.

## Acknowledgements

We would like to thank Rebecca Faill and Jayne Kelleher for technical assistance. *PGC-1β*$^{−/−}$ mice were generated and provided by Transgenic RAD, Discovery Science, Astrazeneca. AV-P was funded by FP7-MITIN (HEALTH-F4-2008-223450) and Medical Research Council Centre for Obesity and Related Metabolic Diseases; SM is funded by a BBSRC grant BB/I020748/1, GN by a BBSRC grant BB/K019260/1 and VIK by BBSRC; SWGT is supported by a Royal Society University Fellowship, GH is supported by a case studentship from BBSRC, CCM is supported by Foundation for Science and Technology (FCT), Portugal studentship through the GABBA Program, University of Porto and Newcastle University, and JFP is supported by a David Phillips Fellowship provided by BBSRC BB/H022384/1 and a BBSRC grant BB/K017314/1.

## Author contributions

CCM did majority of the experiments and contributed to the experimental design and writing of the manuscript; FDMM conducted studies on senescence in *PGC-1β$^{-/-}$ in vitro* and *in vivo* and Rheb overexpression in MEFs and contributed to the experimental design; RA performed IF, mtDNA copy number and Sen-β-Gal activity assays; GH performed Western blots and IF; GN provided technical expertise in microscopy and lentiviral transfection experiments; MC performed T.E.M. in human fibroblasts; AM and SM characterized the effects of rapamycin on mitochondrial function; MDR and DJ conducted real-time PCR; DM contributed to the experimental design; SWGT provided the plasmids and technical support for the YFP-Parkin experiments; AVP and SRC performed experiments using *PGC-1β$^{-/-}$* mice; JB conducted ELISAs; BMC and VIK conducted transfections with mutated RhebN153T and Western blots in liver tissues; MB-Y was involved in the generation of *PGC-1β$^{-/-}$* mice; GQ and DRG performed 3D electron microscopy; LG performed mtDNA copy number assay in human cells. RC and GS conducted TRAP assays; TvZ contributed to the experimental design; JC and PDA performed RNA-seq analysis; RH and PDA conducted oncogene-induced senescence experiments; JFP designed and supervised the study and wrote the manuscript. All authors critically read and commented on the manuscript.

## Conflict of interest

Generation of *PGC-1β$^{-/-}$* mice was funded by AstraZeneca, the employer of Dr. Mohammad Bohlooly-Y. The other authors declare that they have no conflict of interest.

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
