## [Review Process File · The EMBO Journal]

Manuscript EMBO-2015-92862

Mitochondria are required for pro-ageing features of the senescent phenotype

Clara Correia-Melo, Francisco Marques, Rhys Anderson, Graeme Hewitt, Rachael Hewitt, John Cole, Bernadette Carroll, Satomi Miwa, Alina Merz, Michael Rushton, Michelle Charles, Diana Jurk, Stephen Tait, Rafal Czapiewski, Jodie Birch, Laura Greaves, Glyn Nelson, Mohammad Bohlooly-Y, Antonio Vidal-Puig, Sergio Rodriguez-Cuenca, Derek Mann, Gabriele Saretzki, Giovanni Quarato, Douglas Green, Peter Adams, Thomas von Zglinicki, Viktor Korolchuk and Joao Passos

Corresponding author: Joao Passos, Newcastle University

Review timeline:

Submission date:	19 August 2015
Editorial Decision:	23 September 2015
Revision received:	30 November 2015
Accepted:	15 December 2015

Editor: Andrea Leibfried

Transaction Report:

1st Editorial Decision

23 September 2015

Thank you for submitting your manuscript entitled 'Mitochondria are required for pro-ageing features of the senescent phenotype'. I have now received reports from all referees, which are enclosed below.

As you will see, the referees think that your findings are interesting. However, they raise a number of concerns, which can be addressed by further discussion and amendments of the text as well as a few additional experiments. Given the interest in the topic, I would thus like to invite you to submit a revised version of the manuscript to us, addressing all issues raised by the referees.

Importantly,

- All requested controls should be provided (see also point 1 and 7 of referee #2).
- Conflicting results should be discussed (point 6, referee #2), as should be the point made by referee #3 regarding mitochondrial function.
- Referee #1 proposes ways to further support the proposed role of mitochondria in senescence, and including additional experiments as outlined by this referee would in our view further strengthen your manuscript.

Please note that a revised version will be sent back to the same referees. Please contact me in case of questions regarding the revision of your manuscript.

Thank you for the opportunity to consider your work for publication. I look forward to your

revision.

 REFEREE REPORTS

Referee #1:

Cellular senescence accounts for a series of stress events leading to irreversible cell growth arrest. Although senescence is suggested to occur in parallel with mitochondrial dysfunction, an integrative pathway accounting for a mitochondrial induction of senescence remains elusive (Ziegler, AgingCell, 2014). By impinging on parkin-mediated mitophagy or PGC1 modulation of mitochondrial biogenesis, Correia-Melo and cols. report in this work that mitochondrial mass is a key determinant of cellular senescence. The analysis of several markers and inducers of senescence suggest the occurrence of mitochondrial biogenesis during senescence merging an activation of the ATM-Akt-mTORC1- PGC1 axis. Of note, pharmacologically and genetically interfering with this pathway decreases mitochondrial abundance and prevents senescence both in vivo and in vitro, therefore providing a novel approach to counteract senescent damage. The experimental approaches and conclusions are adequate and reading is fluent and clear, as so do diagrams. The commentaries referred below are intended at further implementing and reinforcing the main conclusions of this work.

Main concerns

1. PGC1 data on biogenesis do not report themselves any induction of mtDNA replication. To assess the dependence of the mTORC1-PGC1b axis on mtDNA replication, authors may address the senescent phenotype of mtDNA depleted and/or mtDNA replication deficient cells where mitochondrial biogenesis is interfered. This may also contribute to further elucidate the needs of functional mitochondria to trigger senescence.
2. Despite mitochondrial function is preserved upon mTORC1 inhibition (Fig. 6f), it is not fully clear whether mTORC1 could modulate mitochondrial mass and subsequent changes in senescence markers by means of mitophagy.
3. Changes in mitochondrial mass underlie effects in ROS production. To gain further insight on it, authors are encouraged to specifically address whether ROS responsible for DNA damage in cells with a compromised PGC1 activity have a mitochondrial origin.

Minor concerns

- Sentences before reference to Fig. 2a ("genes down-regulated in Senescence [...]") seem to be repeated and therefore misleading.
- Fig. 2c legend refers to Fig. 5b instead of 2b. For an easier reading, authors are suggested to omit Fig. 2b legend text repeated in methods.
- In the main text, the term "GO" may be specified and long sentences referring to Fig. 2f readdressed for an easier reading. Authors are suggested to evidence the 60% (reversed) or the 5% (exasperated) groups in Fig. 2f.
- Sup. Fig. 2 may not be taken as conclusive of a swift to a glycolytic metabolism (as in the main text) but the full Fig. 2 instead.
- 7 "days" should be specified in Sup. Fig. 3 legend for clearer reading. Sup. Fig. 8i blot may specify Rheb transfection (+ and - signs) and Fig. 5e graph may include "PGC1" in the legend box for easier reading.
- Not being essential for the aim of the figure, inclusion of VDAC/actin data on Sup. Fig. 7a bars should be validated with more than one experiment.
- Fig. 6f is referred to as Fig 7f in the main text.

Referee #2:

The present manuscript from the Passos' lab shows how mitochondria are required for senescence. They use Parkin-mediated mitochondrial clearance as a system to eliminate mitochondria and analyse their role in senescence. They conclude that the absence of mitochondria result in decreased

senescence. Moreover the authors link DNA damage with mitochondrial biogenesis, and describe a pathway involving mTOR, PGC-1 β and ROS production that sustain the senescent phenotype.

In general the paper describes an interesting result. Although links have been previously made between senescence and mitochondria the results presented here deserve to be published. However, there are a few concerns regarding the interpretation of the results and the structure of the manuscript that need to be addressed.

Specific points:

1. The use of Parkin + CCCP is an elegant system to assess the role of mitochondria. However in the experiments presented in Figure 1, some controls are missing. What is the effect of eliminating mitochondria in non-senescent cells? Similarly in the experiments presented in Sup Figures on other types of senescence, the control of treating non-senescent cells with CCCP is missing and necessary to understand the effects of the system completely.
2. In Figure 1h it seems that only 5 factors are induced during irradiation induced senescence. Is this right? Similar results described by the Campisi group (Coppe et al 2008), there are many more secreted factors induced during senescence. Why this discrepancy? If there are only 5 factors induced, it will be simpler to present that data in another form than as a heatmap in which most of the proteins do not change and where there is no sense of significance, error bars or fold induction.
3. Some part of the manuscript are written in a very complicated way. An example is the description of Figure 2 results, which could be simplified.
4. After describing the results in Figure 2, the results drift to describe Sup Figure 2 to 6 for almost 3 pages before mentioning Figure 3a. Sup Figures should be there to support the main Figures and the manuscript. IT will be very difficult to the reader to follow a manuscript that is organised in that way. This needs restructuring, moving essential parts to the main Figures and eliminating parts that are not essential or reorganising them differently.
5. The removal of mitochondria has a very dramatic effect on senescence but it is not clear what happens with the growth arrest. In the irradiation model, it will be better to present stained plates, similar as in Figure 3a to show what is the effect more clearly.
6. The authors show clearly that PGC-1 β is downstream of DDR, however what is the effect of PGC-1 β on DDR. In Fig 3b, there is a decrease of 53BP1 Foci in the KO MEFs, while in Fig 3g there is an increase.
7. All the mechanistic data is produced in human cells except the one regarding PGC-1 β . For consistency it will be better to carry out those experiment in human cells using shRNAs or CrispR.
8. It has been described previously by the Campisi group (Parrinello et al 2003) that MEFs do not undergo senescence at 3% O₂, however the results presented in Fig 3b-d suggest that they do. Can this be explained?
9. The manuscript has 3 very similar schemes at the end of Figures 4, 5 and 6. As the Figures will need to be reorganised I suggest to keep just the last of them. In addition, there are parts of the scheme presented in Figure 6l that have not been shown here (autocrine or paracrine effects of the SASP) and I suggest to take this out of the scheme.

Referee #3:

This is an important and timely manuscript with potential practical implications. The work shows that mitochondrial elimination, via Parkin-mediated ubiquitination and proteolysis, blunts the SASP response (but not the growth arrest response) of cultured cells to radiation. It also shows that mTOR, through induction of the transcriptional co-activator PGC1 β , influences mitochondrial biogenesis and by implication, senescence susceptibility. Although mTOR was already known to affect PGC1 through its influence on the EIF family of mRNA recruitment factors, the manuscript convincingly

addresses ATM and ROS involvement in the mitochondrial biogenesis and senescence responses to DNA damage using an ATM inhibitor, AT fibroblasts, and antioxidants - NAC and Trolox. The sheer volume of data presented is impressive. What is surprising is that despite lower mitochondrial content in the hepatocytes of mice fed rapamycin, mitochondrial function was not significantly altered. It isn't clear how this could be the case - increased efficiency or simply insensitive measurements? The work leaves this and other questions, such as how DNA damage-induced growth arrest is maintained in the absence of a full senescence response, unaddressed. However, the findings presented are still significant, providing further mechanistic rationale for the possible health benefits of mTOR-targeted drugs, as well as identifying additional senescence mediators that may be targeted with greater specificity.

Minor issues to be addressed:

- 1) In Fig. 1c, the left hand panel is missing an arrow corresponding to "oligo".
- 2) In Fig. 1e, it is not clear, which are control and which are Parkin-transduced fibroblasts. Do all the panels presented correspond to Parkin-transduced cells?
- 3) The legend for Figure 2 should read "Note that there is a clear tendency for genes up-regulated in senescence to be down-regulated upon mitochondrial depletion (right of plot),..."
- 4) The text immediately prior to (Figure 2a) should be changed to "genes up-regulated in Senescence to be down-regulated upon mitochondrial depletion."
- 5) In the Figure 2 legend and in the text, the term "exasperated" is used incorrectly, and should be replaced by the term "exacerbated."
- 6) In Fig. 3g, it should be indicated in the panel that this data refers to PGC-1beta overexpression, since the other panels refer to contexts in which the gene is deleted.

1st Revision - authors' response

30 November 2015

Point-by-point response to referees:

We are thankful for the helpful comments and suggestions from the referees. As a result, we believe our manuscript has improved considerably.

Referee #1:

Cellular senescence accounts for a series of stress events leading to irreversible cell growth arrest. Although senescence is suggested to occur in parallel with mitochondrial dysfunction, an integrative pathway accounting for a mitochondrial induction of senescence remains elusive (Ziegler, AgingCell, 2014). By impinging on parkin-mediated mitophagy or PGC1 modulation of mitochondrial biogenesis, Correia-Melo and cols. report in this work that mitochondrial mass is a key determinant of cellular senescence. The analysis of several markers and inducers of senescence suggest the occurrence of mitochondrial biogenesis during senescence merging an activation of the ATM-Akt-mTORC1- PGC1 axis. Of note, pharmacologically and genetically interfering with this pathway decreases mitochondrial abundance and prevents senescence both in vivo and in vitro, therefore providing a novel approach to counteract senescent damage.

The experimental approaches and conclusions are adequate and reading is fluent and clear, as so do diagrams. The commentaries referred below are intended at further implementing and reinforcing the main conclusions of this work.

Main concerns:

1. PGC1 data on biogenesis do not report themselves any induction of mtDNA replication. To assess the dependence of the mTORC1-PGC1b axis on mtDNA replication, authors may address the senescent phenotype of mtDNA depleted and/or mtDNA replication deficient cells where mitochondrial biogenesis is interfered. This may also contribute to further elucidate the needs of functional mitochondria to trigger senescence.

We have shown that liver tissues and MEFs from PGC-1 -/- mice have reduced mtDNA copy number (see Fig EV4B and Appendix Fig S3F). During X-ray induced senescence, mtDNA copy number is increased and this is reduced in PGC-1 -/- MEFs. We have also observed that mTOR inhibition

reduces mtDNA copy number in human fibroblasts during senescence (see Fig 4D) and mouse hepatocytes (see Fig 6E).

As requested by the reviewer, we have now induced IR-induced senescence in Rho(0) cells (where mtDNA depletion was attained by long-term Ethidium bromide exposure). Our results show that ROS is significantly reduced when compared to parental cells, as well as expression of a major SASP component IL-6, further supporting our data that mitochondria impact on the SASP (see Fig EVID). While this experiment further supports our previous data, it is still difficult to draw conclusions about the specific involvement of mtDNA/and mtDNA replication in senescence, particularly since many features of mitochondrial function and composition of the ETC will be equally affected.

2. Despite mitochondrial function is preserved upon mTORC1 inhibition (Fig. 6f), it is not fully clear whether mTORC1 could modulate mitochondrial mass and subsequent changes in senescence markers by means of mitophagy.

This is an excellent point by the reviewer. Similarly to mitochondrial biogenesis, studies have shown that mTORC1 can regulate mitochondrial homeostasis by repressing mitophagy. While we observe evidence for mitochondrial biogenesis upon induction of senescence, our data is not able to discern if effects of mTORC1 on mitochondria are solely due to activation of biogenesis or inhibition of autophagy. 2 pieces of evidence suggest that mTOR driven mitochondrial biogenesis is probably the major contributing factor: i) we observed that upon X-ray irradiation, rapamycin supplementation was still able to decrease mitochondrial mass, ROS and DDR foci in ATG5^{-/-} MEFs (which are devoid of autophagy); ii) Consistent with a major role for mTOR-PGC-1 driving mitochondrial biogenesis in MEFs, we did not observe any additional decrease in mitochondrial mass (measured by MitoTracker green) when we treated PGC-1^{-/-} MEFs with rapamycin (see Figure below). Given the fact that our manuscript has extensive text and supplementary information, we decided not to include the data on autophagy deficient cells in the MS. While our data supports that biogenesis is involved, we do not exclude impairment of autophagy as an alternative process contributing to increased mitochondrial content during senescence and have included a sentence in the discussion acknowledging this possibility.

Figure- mTOR inhibition decreases mitochondrial mass in ATG5^{-/-} cells but do has no effect on PGC-1^{-/-} cells. A) (left) Representative images of LC3 expression in ATG5^{+/+} and ATG5^{-/-} MEFs; (right) representative western blot showing absence of expression of ATG5-12 in ATG5^{-/-} MEFs; B) Effect of 100nM rapamycin on mitochondrial mass (NAO intensity), ROS generation (DHE intensity) and 53BP1 foci number in proliferating and senescent (3 days after 10Gy X-ray) ATG5^{+/+} and ATG5^{-/-} MEFs. Data are mean \pm S.E.M. of n=3-4 independent experiments; C) Effect of 100nM rapamycin on mitochondrial mass (Mito Tracker Green intensity) in proliferating and senescent (3 days after 10Gy X-ray) PGC-1^{+/+} and PGC-1^{-/-} MEFs. Data are from 50-100 cells per condition. Asterisks denote statistical significant P<0.05 using one-way ANOVA.

3. Changes in mitochondrial mass underlie effects in ROS production. To gain further insight on it, authors are encouraged to specifically address whether ROS responsible for DNA damage in cells with a compromised PGC1 activity have a mitochondrial origin.

As asked by referee #2, we stably knocked down PGC-1 in human MRC5 fibroblasts using shRNA (see Fig EV4D). We induced senescence in these cells and measured mitochondrial ROS by measuring MitoSOX intensity (a dye which reacts specifically with superoxide anion inside the mitochondrial matrix). We confirmed that knockdown of PGC-1 in human fibroblasts reduces both basal levels and senescence induced mitochondrial ROS (see Fig EV4E). Further supporting a mitochondrial origin for ROS, we observed that cellular superoxide anion levels (measured with DHE) and peroxides (measured using DHR123-not shown) were decreased to basal levels following senescence induction upon Parkin-mediated mitochondrial clearance (see Fig 1F).

Minor concerns:

- Sentences before reference to Fig. 2a ("genes down-regulated in Senescence [...]") seem to be repeated and therefore misleading.

We apologise for the mistake, we have corrected it.

- Fig. 2c legend refers to Fig. 5b instead of 2b. For an easier reading, authors are suggested to omit Fig. 2b legend text repeated in methods.

We have corrected it.

- In the main text, the term "GO" may be specified and long sentences referring to Fig. 2f readdressed for an easier reading. Authors are suggested to evidence the 60% (reversed) or the 5% (exasperated) groups in Fig. 2f.

We have corrected it. As suggested by the reviewer we have evidenced in Figure 2 the genes which are reversed (coloured in green) and the exacerbated ones (coloured in purple).

- Sup. Fig. 2 may not be taken as conclusive of a swift to a glycolytic metabolism (as in the main text) but the full Fig. 2 instead.

We clarified the results and conclusions in the text. We believe our data does support a shift to glycolysis. In Supplementary Fig.2 (now Figure EV2) we show that in terms of ECAR, Parkin+CCCP have a high basal rate, which does not respond to inhibition of OXPHOS ATP synthesis (whereas in other cells, we see compensatory upregulation of ECAR after OXPHOS ATP inhibition). Addition of 2DG (which stops glucose transport to cells, commonly used inhibitor for glycolysis) has a very strong effect on Parkin CCCP when compared to Parkin DMSO. We have now replotted the graphs so that the data is more easily interpreted (see Figure EV2A). In our view, these data, together with the RNA-seq data demonstrating increased mRNA expression of glycolytic genes supports a shift to glycolysis.

- 7 "days" should be specified in Sup. Fig. 3 legend for clearer reading. Sup. Fig. 8i blot may specify Rheb transfection (+ and - signs) and Fig. 5e graph may include "PGC1" in the legend box for easier reading.

We have corrected it.

- Not being essential for the aim of the figure, inclusion of VDAC/actin data on Sup. Fig. 7a bars should be validated with more than one experiment.

We have excluded the western blot for VDAC. We have extensively characterised several other mitochondrial proteins, so we agree it is not necessary to include it.

- Fig. 6f is referred to as Fig 7f in the main text.

We apologise for the mistake, we have corrected it.

Referee #2:

The present manuscript from the Passos' lab shows how mitochondria are required for senescence. They use Parkin-mediated mitochondrial clearance as a system to eliminate mitochondria and analyse their role in senescence. They conclude that the absence of mitochondria result in decreased senescence. Moreover the authors link DNA damage with mitochondrial biogenesis, and describe a pathway involving mTOR, PGC-1b and ROs production that sustain the senescent phenotype.

In general the paper describes an interesting result. Although links have been previously made between senescence and mitochondria the results presented here deserve to be published. However, there are a few concerns regarding the interpretation of the results and the structure of the manuscript that need to be addressed.

Specific points:

1. The use of Parkin + CCCP is an elegant system to assess the role of mitochondria. However in the experiments presented in Figure 1, some controls are missing. What is the effect of eliminating

mitochondria in non-senescent cells? Similarly in the experiments presented in Sup Figures on other types of senescence, the control of treating non-senescent cells with CCCP is missing and necessary to understand the effects of the system completely.

As requested by the referee, we have now included results showing that mitochondrial clearance in young/proliferating cells does not induce senescence-associated phenotypes such as Sen- -Gal activity and over-production of pro-oxidant and pro-inflammatory signals. Moreover, these cells show no compromised ATP production. However, elimination of mitochondria in young cells resulted in a significant decrease of cell proliferation (see new Figures EV1G-I). One possibility to explain the latter phenotype is that that clearance of mitochondria reduced significantly mTOR activity (see new Figures EV1F and EV1G). Consistent with this hypothesis and our own data, are recent studies which have shown that the mechanisms by which mTOR impacts on the cell-cycle during senescence are uncoupled from its effect on the SASP (Laberge et al. & Herranz et al. 2015). However, we acknowledge that it is possible that mitochondria may impact on cell division independently from mTOR.

We agree with the reviewer that the role of mitochondria in normal proliferating cells is an important question and we expect that lack of mitochondria will have a multitude of effects on cells (we are presently investigating these effects and hope to publish in future studies). However, we believe that this is not in the scope of the present MS, where we intended to test the necessity of mitochondria for the development of cellular senescence.

2. In Figure 1h it seems that only 5 factors are induced during irradiation induced senescence. Is this right? Similar results described by the Campisi group (Coppe et al 2008), there are many more secreted factors induced during senescence. Why this discrepancy? If there are only 5 factors induced, it will be simpler to present that data in another form than as a heatmap in which most of the proteins do not change and where there is no sense of significance, error bars or fold induction.

In the abovementioned study Coppe and colleagues have used different fibroblast strains from MRC5 fibroblasts. While there are many similarities in human fibroblasts strains in terms of SASP components, we and others have noted that the extent of secretion of SASP components varies considerably between fibroblast cell lines. We have consistently only detected the described 5 SASP factors (above detection level) in the media of senescent MRC5 fibroblast using a Raybiotech antibody array. It is possible that other SASP factors are increased (as the mRNA profile suggests) but their detection requires more sensitive methods. As advised by the reviewer, we have replaced the heatmap by individual graphs for the SASP factors that show a robust induction and performed statistical analysis.

3. Some part of the manuscript are written in a very complicated way. An example is the description of Figure 2 results, which could be simplified.

We appreciate the comment and have simplified the manuscript, particularly the description of Figure 2.

4. After describing the results in Figure 2, the results drift to describe Sup Figure 2 to 6 for almost 3 pages before mentioning Figure 3a. Sup Figures should be there to support the main Figures and the manuscript. IT will be very difficult to the reader to follow a manuscript that is organised in that way. This needs restructuring, moving essential parts to the main Figures and eliminating parts that are not essential or reorganising them differently.

We thank the reviewer for the suggestions and have reorganised the manuscript text and figures (see new Figure 2, EV2 and EV3). We believe the manuscript is now clearer and the message easier to follow.

5. The removal of mitochondria has a very dramatic effect on senescence but it is not clear what happens with the growth arrest. In the irradiation model, it will be better to present stained plates, similar as in Figure 3a to show what is the effect more clearly.

We have attempted to conduct the requested experiment, however there are some technical issues: we have noticed that following mitochondrial clearance, Parkin-expressing human fibroblasts need

to be seeded above a certain density (we have included a more detailed description of how to culture these cells in material and methods) and that isolated single cells detached easily, which precludes the execution of colony assays. However, we have performed BrdU and/or EdU incorporation for all the models of senescence (see Figure EV3), which match with quantification of cell numbers.

6. The authors show clearly that PGC-1b is downstream of DDR, however what is the effect of PGC-1b on DDR. In Fig 3b, there is a decrease of 53BP1 foci in the KO MEFs, while in Fig 3g there is an increase.

Fig 3G (now Fig 3F) refers to overexpression of PGC-1 in MEFs, where in contrast to PGC-1 -/- MEFs we observe a further increase of DDR foci. We apologise that this was not clear in the figure (as also noted by referee 3); we now made sure that this was clearly labelled. Our data supports the idea that increased expression of PGC-1 increases cellular ROS and this contributes to an enhanced DDR. Consistent with this hypothesis: i) absence of PGC-1 reduces both ROS and DDR (see Figure 5D and 5F) and ii) antioxidant supplementation prevents the increase of the DDR driven by overexpression of PGC-1 in MEFs (see Figure 5E).

7. All the mechanistic data is produced in human cells except the one regarding PGC-1b. For consistency it will be better to carry out those experiments in human cells using shRNAs or Crispr.

As requested by the reviewer we have conducted lentiviral mediated shRNA against PGC-1 in human fibroblasts (knock-down was confirmed by western blot and RT-PCR). Upon induction of senescence, and similarly to what we observed in PGC-1 -/- MEFs, we observed reduced mitochondrial mass, superoxide anion levels, 53BP1 foci and frequencies of senescent marker Sen- Gal. However, we did not observe a proliferation rescue (as measured by EdU incorporation) as observed in MEFs (see Figure EV4D-F). A possible explanation may derive from the fact that the effects of PGC-1 on mitochondrial biogenesis are more acute in MEFs than in human fibroblasts (60% knock-down of PGC-1 only reduced mitochondrial mass induction in Sen (IR) by 25%).

8. It has been described previously by the Campisi group (Parrinello et al 2003) that MEFs do not undergo senescence at 3% O₂, however the results presented in Fig 3b-d suggest that they do. Can this be explained?

We believe there is no contradiction between our data and that of the Campisi lab: similarly to Parrinello et al. 2003 we observe that MEFs cultured at 3% oxygen do not become senescent, however, upon X-ray irradiation, MEFs can acquire a senescent phenotype which is stabilised by positive feedback loops involving ROS and the DDR. Consistently, the paper by Parrinello and colleagues shows that MEFs growing at 3% oxygen retain DNA damage checkpoints. Furthermore, similarly to our results, the same group have reported that at 3% oxygen, MEFs become senescent 10 days after 10Gy X-ray irradiation (with less than 5% BrdU incorporation >70% Sen- Gal and a robust SASP) (Coppe et al. 2010). We realised that the figure was not correctly labelled- we clarified in Figure 3 that when MEFs were cultured at 3%O₂, senescence was induced by X-ray irradiation (Sen IR).

9. The manuscript has 3 very similar schemes at the end of Figures 4, 5 and 6. As the Figures will need to be reorganised I suggest to keep just the last of them. In addition, there are parts of the scheme presented in Figure 6l that have not been shown here (autocrine or paracrine effects of the SASP) and I suggest to take this out of the scheme.

We agree with the reviewer that having a summary scheme for each figure may become confusing and as advised we have removed them leaving just the last one in Figure 6.

Referee #3:

This is an important and timely manuscript with potential practical implications. The work shows that mitochondrial elimination, via Parkin-mediated ubiquitination and proteolysis, blunts the SASP response (but not the growth arrest response) of cultured cells to radiation. It also shows that mTOR, through induction of the transcriptional co-activator PGC1beta, influences mitochondrial biogenesis and by implication, senescence susceptibility. Although mTOR was already known to affect PGC1

through its influence on the EIF family of mRNA recruitment factors, the manuscript convincingly addresses ATM and ROS involvement in the mitochondrial biogenesis and senescence responses to DNA damage using an ATM inhibitor, AT fibroblasts, and antioxidants - NAC and Trolox. The sheer volume of data presented is impressive. What is surprising is that despite lower mitochondrial content in the hepatocytes of mice fed rapamycin, mitochondrial function was not significantly altered. It isn't clear how this could be the case - increased efficiency or simply insensitive measurements? The work leaves this and other questions, such as how DNA damage-induced growth arrest is maintained in the absence of a full senescence response, unaddressed. However, the findings presented are still significant, providing further mechanistic rationale for the possible health benefits of mTOR-targeted drugs, as well as identifying additional senescence mediators that may be targeted with greater specificity.

We appreciate the comments from the referee. We completely agree that there are still a number of unanswered questions arising from this work, but strongly feel that these should be addressed in future studies given the complexity and volume of data presented in the current manuscript. Our finding that mitochondrial function is not significantly altered in mice fed with rapamycin has also been reported independently (see Johnson et al. 2013, Miwa et al. 2014), despite changes in the composition of ETC complexes and reduced ROS generation (Miwa et al. 2014). As our mitochondrial function measurements were conducted in isolated mitochondria (we have clarified it in the figure legend and text), we don't believe there is a direct contradiction between having low mitochondrial content per cell and no change in function per unit of mitochondria. However, we acknowledge that it remains a possibility that the method used is insensitive to small differences particularly considering the extensive mouse-to-mouse heterogeneity and cannot exclude from our data that several other mTOR-dependent effects apart from mitochondrial content may impact on senescence in vivo.

We agree with the referee that the uncoupling between the cell-cycle arrest and the SASP is an important question. It is possible that mitochondria may be involved directly in processes mediating the cell-cycle; whether by generation of growth-stimulating metabolites or interaction with cytoplasmic elements required for cell division as previous work as demonstrated. Our data also points out that mitochondria clearance reduces mTOR activity, which by itself inhibits both cell proliferation and the SASP. We have included an additional paragraph in the discussion to address this particular issue.

Minor issues to be addressed:

1) In Fig. 1c, the left hand panel is missing an arrow corresponding to "oligo".

We have corrected it.

2) In Fig. 1e, it is not clear, which are control and which are Parkin-transduced fibroblasts. Do all the panels presented correspond to Parkin-transduced cells?

We apologise for not making it clear in the figure. We have now included a title in Figure 1e describing that the panel corresponds to Parkin transduced cells only.

3) The legend for Figure 2 should read "Note that there is a clear tendency for genes up-regulated in senescence to be down-regulated upon mitochondrial depletion (right of plot),..."

We have corrected it.

4) The text immediately prior to (Figure 2a) should be changed to "genes up-regulated in Senescence to be down-regulated upon mitochondrial depletion."

We have corrected it.

5) In the Figure 2 legend and in the text, the term "exasperated" is used incorrectly, and should be replaced by the term "exacerbated."

We have corrected it.

6) In Fig. 3g, it should be indicated in the panel that this data refers to PGC-1beta overexpression, since the other panels refer to contexts in which the gene is deleted.

We apologise for this not being clear and have corrected it.

2nd Editorial Decision

15 December 2015

Thank you for submitting your revised manuscript to us. It has now been seen by two of the original referees whose comments are enclosed. As you will see, both referees appreciate the introduced changes. I am thus happy to accept your manuscript in principle for publication in The EMBO Journal.